# A behavioral approach to instability pathways in financial markets

Alessandro Spelta[1,2 ✉], Andrea Flori [3], Nicolò Pecora[4], Sergey Buldyrev[5] & Fabio Pammolli[2,3]

We introduce an indicator that aims to detect the emergence of market instabilities by quantifying the intensity of self-organizing processes arising from stock returns' co-movements. In financial markets, phenomena like imitation, herding and positive feedbacks characterize the emergence of endogenous instabilities, which can modify the qualitative and quantitative behavior of the underlying system. The impossibility to formalize ex-ante the dynamic laws that rule the evolution of financial systems motivates the use of a parsimonious synthetic indicator to detect the disruption of an existing equilibrium configuration. Here we show that the emergence of an interconnected sub-graph of stock returns co-movements from a broader market index is a signal of an out-of-equilibrium transition of the underlying system. To test the validity of our approach, we propose a model-free application that builds on the identification of up and down market phases.

[1] Department of Economics and Management, University of Pavia, Via San Felice 7, 27100 Pavia, Italy. [2] CADS, Joint Center for Analysis, Decisions and Society, Human Technopole, Milan, Italy. [3] Department of Management, Economics and Industrial Engineering, Politecnico di Milano, Via Lambruschini, 4/B, 20156 Milan, Italy. [4] Department of Economics and Social Sciences, Catholic University, Via Emilia Parmense 84, 29122 Piacenza, Italy. [5] Department of Physics, Yeshiva University, 500 West 185th Street, Belfer Hall, New York City, NY, USA. ✉email: alessandro.spelta@unipv.it

This paper tackles the issue of detecting long-range memory and co-movements across financial time series as informative signals of market instability and of upcoming changes in the dynamic laws governing the evolution of the system. A precise mathematical description of the underlying system through dynamic equations is, in fact, not feasible during the transition between equilibria. An in-depth inspection of the system is thus instrumental to uncover how the evolving relationships between market participants induce distinguishable variations in the set of financial variables, which may lead to instability[1–5].

Imitation, herding behaviors, and positive feedbacks among market participants have been recognized as phenomena leading to endogenous instabilities[6,7]. Herding behaviors spread when the knowledge about other investors' allocation decisions influences personal strategies, meaning that investors tend to use similar investment practices to those applied by other market participants even when this is not justified by their own information set[8–10], while positive feedbacks can induce the underlying system to accumulate instabilities that lead to new configurations as a self-fulfilling mechanism[11–14]. Hence, a strengthening of interactions among asset prices may emerge due to market euphoria, which drives prices to a sharp rise or, by contrast, to phenomena of financial turmoil, which induce fire sales and stock market crashes.

Several techniques have been applied in the literature to study how cross-market linkages, co-movements, and interdependencies between stocks contribute to influence the sustainability conditions of financial markets and, possibly, the mechanisms behind shock transmission[15–19]. Here, against this background, we focus on the intensity of self-organizing processes arising from stock returns' co-movements and self-similarities.

Inspired by H.A. Simon's near decomposability condition[20] to represent a stable system configuration[21–23], we hypothesize that, during instability phases, a sub-graph of stocks displays increasing co-movements and self-similarity patterns, which we propose to quantify by means of the Pearson's correlation coefficient (PCC) and the autocovariance (AC) of stock returns (see Supplementary information, Section 3.1). We refer to this sub-graph of stocks as the leading temporal module (LTM) of the system, whose dynamics is anticipatory for the whole underlying system. In particular, when the system is approaching a change in its equilibrium configuration, we observe that the absolute value of the average PCC is increasing within the set of stocks composing the LTM sub-graph, but decreasing between stocks belonging to the LTM and stocks outside the LTM group, while the average AC of stocks within the LTM is increasing.

A rigorous investigation of the properties of the LTM, based on its temporal and spatial dimensions, allows us to build a synthetic and flexible indicator, which we use to detect the emergence of significant changes in the underlying financial market. We propose a parsimonious aggregate indicator based on the mean absolute value of the AC of the stocks belonging to the LTM ($<|AC_t^{LTM}|>$) and on the ratio between the correlations of stocks within the LTM ($<|PCC_t^{LTM}|>$) and the correlations of stocks outside the leading module ($<|\widetilde{PCC_t^{LTM}}|>$). We relate the first component to the existence of positive feedbacks in the market[24,25], while the second component reveals the presence of herding behaviors among investors[26,27]. The corresponding synthetic indicator is defined accordingly as:

$$I_t^{LTM} = \frac{<|AC_t^{LTM}|><|PCC_t^{LTM}|>}{<|\widetilde{PCC_t^{LTM}}|>}. \tag{1}$$

To identify those stocks that have a higher potential triggering, before applying the LTM procedure, we use the detrended

fluctuation analysis (DFA)[28–30] on the time series of the original returns and on data obtained by independent time permutation. We focus on stocks that show DFA exponents significantly different from 1/2, which is the expected value for a memoryless signal. Hence, the LTM identification is performed within the set of stocks for which the DFA indicates the presence of long-range memory. For comparison purposes, we also verify the predictive properties of both the set of stocks that have a statistically significant DFA, but are not in the LTM sub-graph (namely, DFA⁻) and the ones not selected by neither the DFA nor the LTM procedures (indicated as Rest).

To mimic the possible system dynamics far and near a transition point, we also employ a Lotka–Volterra model of stocks dynamics (see Supplementary information, Section 3.3). Specifically, we simulate the system with different values of the bifurcation parameter and then we compute the statistical components of our proposed indicator. We note that while far from transition the time series exhibit small correlations and relatively low ACs, close to the bifurcation point the series exhibit both higher ACs and stronger correlation values.

Finally, we implement an illustrative investment strategy that builds on the identification of the emergence of up and down market phases[31,32] to show the functioning of our approach. Our analysis thus contributes to the understanding of financial markets by studying how the effects of linkages at the micro-level turn out to be relevant at the macro-level in the corresponding aggregate system. In fact, at the micro-level investors interact through heterogeneous allocation strategies, adapting their behavior in response to the performance of their investments, the arrival of new information, and the interplay of social interactions and observations, which generate, at the macro-level, non-trivial aggregate patterns of the corresponding financial system[13,33–40]. Related to our work is, therefore, the approach of employing community detection methodologies[41–43] to understand the properties of the dynamic processes taking place in a correlation network, from which the detection of the LTM is inspired.

Moreover, we can establish a link between our approach and what is observed in natural sciences, since variations in asset prices can be seen as the social equivalent of nucleation phenomena near the limit of stability in a thermodynamic system, such as a superheated liquid or supercooled gas[44]. In our approach, the LTM can be viewed as analogous to the nucleus of the new phase for financial markets. We can say the indicator $I_t^{LTM}$ plays a role similar to compressibility in thermodynamic systems, that is, the macroscopic thermodynamic quantity referring to the increasing instability near the spinodal lines.

## Results

**The LTM indicator**. We analyze the stocks referring to the STOXX Asia/Pacific 600 Index, which is a broad and liquid subset of the STOXX Global 1800 Index. We investigate the dynamics of the aggregate index starting from the micro-level represented by the stocks that approximately constitute it. We employ daily closure prices along the period 2006–2017 to compute the corresponding returns at the ground of the analysis. During the period of our analysis, the Asian stock market experienced unstable dynamics, with large booms and bursts. These up and down swings reflect the 2008 global financial crisis firstly, and, more recently, the real estate bubble and the flood of debt by municipal governments and local enterprises designed to fund infrastructure investments[45–48]. We also provide additional evidence on stocks constituting the STOXX North America 600 Index (see Supplementary information, Section 3.7).

The dynamics of the LTM sub-graph identifies market phases characterized by the strengthening of price co-movements

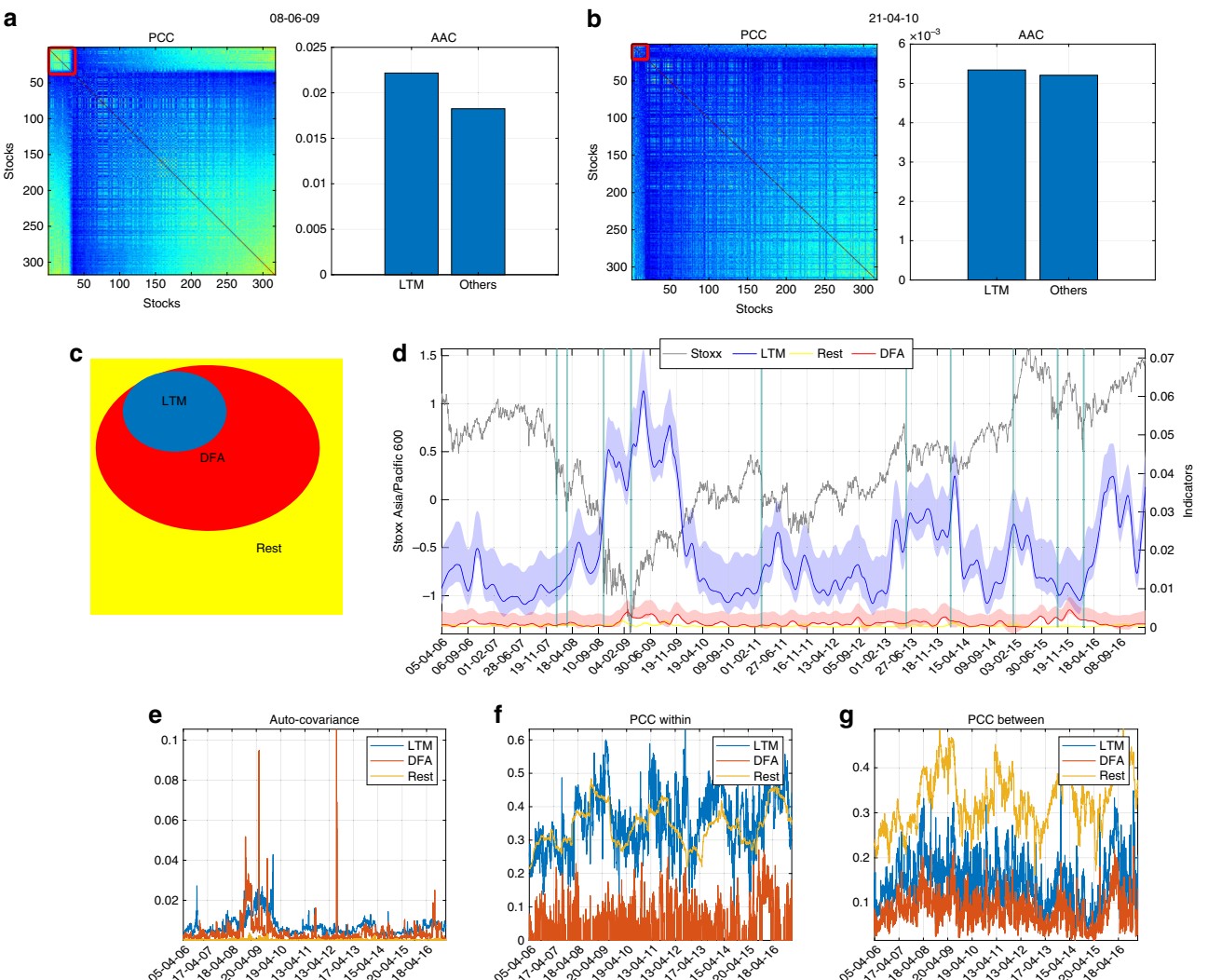

**Fig. 1 The leading temporal module (LTM) sub-graph in different market phases together with the indicator $I_t^{LTM}$ reported against the market dynamics. a**, **b** show the absolute value of the correlation matrices (PCC) derived from stocks returns, emphasizing the LTM sub-graph with a red square, together with the absolute average auto-covariance (AAC) computed on both its members and on the rest of the system. Correlation matrices and auto-covariance are displayed for two different market phases, centered around 08-06-2009 in **a** and around 21-04-2010 in **b**, which stand for an unstable and a business as usual phase, respectively. **c** illustrates the sets of stocks within the system: stocks composing the LTM, stocks that have a statistically significant DFA but are not in the LTM sub-graph (DFA⁻), and the rest of the stocks not selected by neither the LTM algorithm nor the DFA (Rest). **d** reports the leading indicator (right axis) computed on the LTM members (blue line), on DFA⁻ members (red line), and on the Rest of the stocks (yellow line). These indicators have been smoothed based on a Lowess (locally weighted scatter-plot smoothing) filter and compared with the dynamics of the underlying reference index displayed in gray (left axis). Vertical bars correspond to crisis events affecting the financial market such as the banking sector ratings downgrades of 2007, the failure of Lehman Brothers in September 2008, the American Recovery and Reinvestment Act of 2009, the European Debt crisis of 2011, and the Chinese stock market crisis of 2015–2016. Error bounds are computed by performing 500 bootstrapping re-sampling of stocks' returns from the empirical distribution of the observed data and computing for each run the LTM indicator. Shaded areas represent the 5–95% confidence intervals. Two-sample Kolmogorov–Smirnov (KS) test provides evidence about the statistical difference between $I^{LTM}$ and the indicators computed on DFA⁻ and Rest. The pairwise KS statistics of $I^{LTM}$ vs. the indicators for DFA⁻ and Rest are 0.95 and 0.99 at 1% significance level, respectively, thus suggesting that they are from different continuous distributions. **e–g** show the dynamics of the components of the leading indicator; from the left to right: the absolute auto-covariance of stocks' returns (**e**), the within cluster absolute Pearson's correlation (**f**) and the between clusters absolute Pearson's correlation (**g**). All computations are made using a moving window of 200 days.

responsible for the transition of the whole underlying market away from its current configuration. Upper panels of Fig. 1 show the absolute values of both the correlation and the AC for the LTM members and for the other stocks not included in the LTM. Figure 1a refers to an unstable period (centered around 08-06-2009), while Fig. 1b refers to a stable phase (centered around 21-04-2010). Figure 1c shows the schematic diagram of the sets of stocks composing the system: the LTM group (labeled as LTM), the stocks that have a statistically significant DFA, but

that are not in the LTM sub-graph (labeled as DFA⁻), and the rest of the stocks not selected by neither the LTM algorithm nor the DFA (labeled as Rest). Figure 1d shows the dynamics of the underlying index (in gray), the pattern of $I_t^{LTM}$ referring to the LTM members (blue line) and the analogous indicators computed on stocks belonging to the DFA⁻ group (red line) and on the Rest group (yellow line). Figure 1 shows how, during unstable phases, the LTM emerges in the correlation matrix, displaying also relatively high values of the ACs (Fig. 1a). On the

contrary, the module is indistinguishable from the remaining part of the system during "business as usual" phases (Fig. 1b). In a nutshell, $I_t^{LTM}$ increases and assumes higher values around periods of market instability than during a tranquil period. For instance, during the 2008 global financial crisis, $I_t^{LTM}$ starts to increase prior to the outbreak of the market and it reaches a local maximum approximately in correspondence of the onset of the crisis. Figure 1d also points out that $I_t^{LTM}$ shows an increasing dynamics in correspondence of major events affecting the market, such as the banking sector ratings downgrades of 2007, the failure of Lehman Brothers in September 2008, the American Recovery and Reinvestment Act of 2009, the European Debt crisis of 2011, and the Chinese stock market crisis of 2015–2016. Moreover, $I_t^{LTM}$ shows a sharp increase also for transitions occurring during positive market trends, as for instance in the recovery period after the global financial crisis and at the end of the sample period. By contrast, the dynamics of both DFA$^-$ and the Rest groups seem less informative in distinguishing phases of instability in the market as reported in Fig. 1d. Error bounds are computed by performing 500 bootstraps re-sampling by randomly permuting stocks' returns. Every bootstrap sample allows acquiring an estimate of $I_t^{LTM}$, which is used to compute the distribution of the indicator and to estimate the error bound as the 5–95th percentiles of such distribution.

Our analysis of the system at different points in time is able to identify stages of accumulation of market instability by detecting qualitative changes in the structure of the interactions among market participants.

The dynamics of the LTM mimics some behavioral attitudes of market participants, such as positive feedbacks and herding behaviors that reverberate in the path of stock prices. The former empirically translates into an increased AC of stock returns, while the latter empirically drives an increase of the correlation of such returns. The lower panels of Fig. 1 (panels e, f, and g) show the time dynamics of the components of $I_t^{LTM}$ as described by Eq. (1). From the left to the right: the absolute AC of stocks' returns (panel e), the within-group absolute Pearson's correlation (panel f), and the between groups absolute Pearson's correlation (panel g). These components jointly contribute to detect the emergence of phases of cumulative market instabilities. In particular, the AC signals the presence of positive feedbacks around the outbreak

of the global financial crisis and of its rebound, while high correlation values between the LTM members indicate the presence of a bunch of stocks having strong synchronized patterns, which deviate from the behavior of the rest of the system. Notice that when these components are evaluated separately, they do not provide a clear interpretation of market conditions, while only once jointly considered they convey a meaningful signal.

$I_t^{LTM}$ is a dynamic indicator whose members may vary in time. Changes in the LTM composition are important to identify the drivers of the upcoming period of instability. In order to investigate the composition of the LTM sub-graph, its size and the entry–exit dynamics of the stocks in the module, we report, in Fig. 2a, the stability coefficient of the LTM computed as the portion of stocks that belong to the module during two consecutive days (green line). We also report the size of the LTM (in red) and that of the group DFA$^-$ (in blue), expressed as percentages to the total number of stocks composing the reference index. In the lower part of Fig. 2a, we also report the correlation between the number of stocks selected by the DFA procedure and the average correlation of these stocks' returns. This helps us to verify whether a rise in the number of stocks with a significant DFA exponent is related to the growth of the average correlation of the returns associated with these stocks, and thus to a higher likelihood of being LTM members. When most of the stocks with a significant DFA exponent belong to the LTM (see red line), we observe a stable dynamics of the leading module (see green line) or, to put it differently, a low turnover of the stocks inside the LTM. On the contrary, the LTM stability drastically decreases when there exists a considerable amount of stocks with a significant DFA exponent that are not part of the leading module (see blue line). Indeed, we observe positive and high values of the correlation when most of the stocks selected by the DFA also belong to the LTM as, for instance, during the 2008 global financial crises and during the last semester of 2015, after the Renmimbi devaluation, while low values of the correlation are associated with periods of substantial changes of the LTM members. The negative Pearson's correlation ($-0.19$) between the LTM stability coefficient and the size of the subset of stocks composing the DFA$^-$ group (i.e., not included in the leading module) indicates that new leading modules are likely to emerge in those periods in which there are stocks with significant DFA exponents but with poorly correlated returns. In the Section 3.4 of

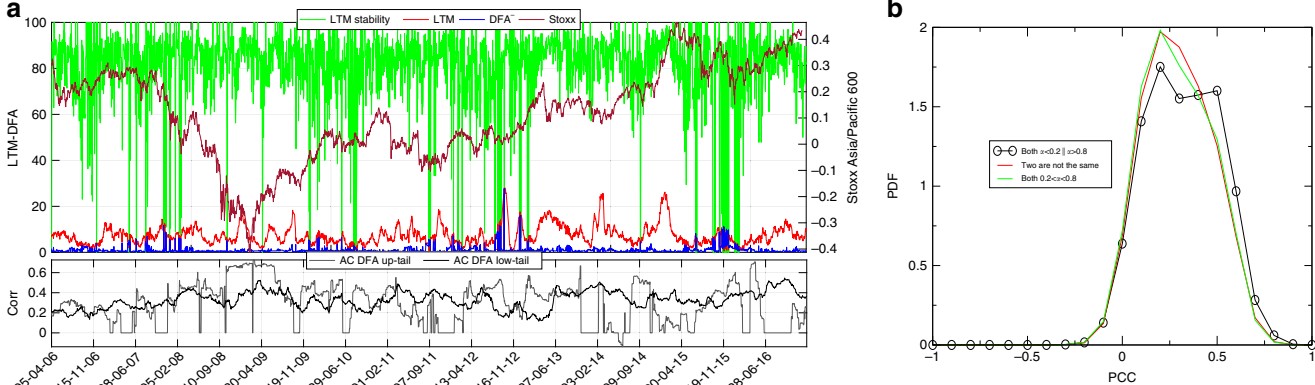

**Fig. 2 LTM membership stability. a** Shows in green the percentage of stocks that belong to the LTM for two consecutive trading days (LTM stability), while the red line stands for the percentage of stocks belonging to the LTM. The blue line emphasizes the percentage of DFA$^-$ stocks. The black line shows the value of the correlation (Corr) between the number of stocks selected by the DFA and the average correlation of these stocks' returns, for those having a DFA exponent significantly larger than 1/2 (AC DFA up-tail), while the gray line refers to stocks with a DFA exponent significantly lower than 1/2 (AC DFA low-tail). Finally, the dynamics of the series are compared with the market index (reported in orange). **b** Shows the distributions of pairs of stocks such that both stocks have Hurst exponent outside the interval 0.2–0.8 (in black), of pairs such that none of them is outside the interval 0.2–0.8 (in green) and those with only one stock belonging to such interval (in red).

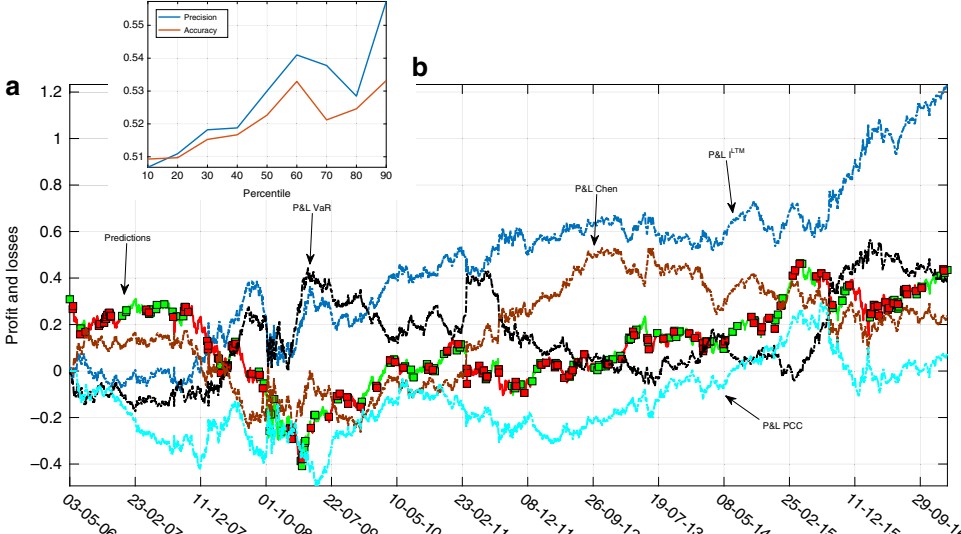

**Fig. 3 The buy/sell signals provided by the dynamics of $I_t^{\text{LTM}}$ and of its members' returns together with the obtained Profits and Losses (P&L).** The figure reports, in **a**, the forecast dynamics of the benchmark index (green-red colors stand for buy and sell signals, respectively) together with the P&L of the investment strategy based on $I_t^{\text{LTM}}$ (blue line). The P&L of an investment strategy based on the value at risk (VaR), that is, the maximum potential loss computed on a daily time horizon with an interval of confidence of 0.975 is also reported (black line) as a comparative measure. The true positives, false positives, false negatives, and true negatives obtained by investing following $I_t^{\text{LTM}}$ are 53%, 47%, 49%, and 51%, respectively, while for the strategy based on the VaR, we obtain 49%, 51%, 52%, and 48%. Finally, the brown line refers to the P&L evolution obtained by considering an investment strategy based on the indicator proposed by ref. [58], while the cyan line shows the P&L obtained when considering only the average correlation among stocks' returns. In **b**, we report the accuracy and precision measures of the proposed investment strategy conditionally on the forecast of the market index returns larger than a certain percentile of their distribution in absolute terms.

the Supplementary information, we also report that, on average, stocks stay continuously in the leading module for about 1.5 months. Figure 2b shows the distributions of the correlations between pairs of stocks such that both stocks have Hurst exponent outside the interval 0.2–0.8 (in black) in comparison with the pairs such that none of them is outside that interval (in green) or such that only one of the stock in the pair belongs to that interval (in red). The shifting to the right of the distribution for the stocks with Hurst exponent outside the interval 0.2–0.8 suggests that the DFA selects assets with high correlated returns. In other words, an increase of the number of stocks with a significant DFA exponent implies an increase of the average correlation of the returns associated with these stocks and subsequently to a higher probability of entering the LTM.

**The predictive performance.** We assess the predictive performance of $I_t^{\text{LTM}}$ by testing an investment strategy that consists of two steps: first, the detection of a cumulative process leading to a phase of instability and, second, the identification of the market direction. As for the first point, a thresholding approach for extracting signals from $I_t^{\text{LTM}}$ appears not suitable, since it would require the prior knowledge of the out-of-sample distribution. Therefore, the most recent value of the indicator is compared against its empirical distribution computed over the previous three trading weeks (15 working days). If this value belongs to the right tail of the distribution, then the LTM is interpreted as signaling a cumulative process leading to market instability. Instead, to detect the direction of the market trend, we exploit the most recent returns of the LTM members, averaging among such values at the day corresponding to the investment decision. If the average value of the returns is positive, then the signal conveyed by the LTM indicates the arrival of a shift towards a bullish equilibrium; otherwise, it stands for a declining and bearish

market dynamics. In other words, whenever $I_t^{\text{LTM}}$ falls in the right tail of its empirical distribution, a switch between a long or short (or vice versa) investment position is possible depending on the average returns of the LTM members. More specifically, we refer to values of the $I_t^{\text{LTM}}$ larger than the 95th percentile of its empirical distribution as the trigger for switching the investment exposure: if the average return of the stocks composing the LTM sub-graph is positive at time $t$, then we opt for a buy signal in that day; otherwise, the strategy goes short.

Figure 3a reports the behavior of the underlying market index in which price forecasts are emphasized by green (i.e., buy) and red (i.e., sell) colors. Notice how, when there is a declining dynamics of the market index, our strategy mostly signals a short position, while in ascending price phases the green color prevails, indicating a long portfolio exposure. In particular, prior to the global financial of 2008, the indicator is able to correctly anticipate the price downturn, while at the onset of the crisis the wave of financial turbulence prevents a clear market trend detection. However, the subsequent rebound is timely identified. Figure 3a also shows the Profit and Loss (P&L) of the strategy based on $I_t^{\text{LTM}}$ (in blue) to disentangle the phases in which up-down movements allow to obtain positive portfolio performances. In fact, the proposed strategy is able to generate a positive cumulative performance along the sample period. In Fig. 3a, we also compare the P&L of an investment strategy based on a well-known measure of risk such as the value at risk (VaR)[49,50], which estimates the maximum amount of expected loss over a specified time horizon at a given confidence level (see Supplementary information, Section 3.5). High values of VaR, that is, values higher than the 95th percentile of its empirical distribution, suggest a phase of instability, and, accordingly, the strategy takes a short position; otherwise, it goes long. Notice how, while $I^{\text{LTM}}$ recognizes changes in market trajectories in a timely way, the investment strategy based on VaR (in black), on the other hand, is

**Table 1 Profits and Losses (P&L) performances.**

|  | 2006 | 2007 | 2008 | 2009 | 2010 | 2011 | 2012 | 2013 | 2014 | 2015 | 2016 | 2017 | 2006–17 |
|---|---|---|---|---|---|---|---|---|---|---|---|---|---|
| MV = 10; PRCTILE = 90 | 3.15 | −12.80 | 7.90 | 11.96 | 11.64 | −2.04 | 2.48 | 2.96 | 3.78 | 19.78 | 2.50 | 4.76 | 56.05 |
| MV = 10; PRCTILE = 95 | 2.27 | 6.42 | −8.48 | 20.26 | 3.76 | −7.51 | 19.82 | 15.40 | 1.32 | 29.54 | 25.48 | 6.16 | 114.44 |
| MV = 10; PRCTILE = 98 | 2.27 | 6.42 | −8.48 | 20.26 | 3.76 | −7.51 | 19.82 | 15.40 | 1.32 | 29.54 | 25.48 | 6.16 | 114.44 |
| MV = 15; PRCTILE = 90 | 6.86 | −16.37 | 24.59 | 14.00 | 25.96 | −3.23 | 11.21 | −17.76 | −6.19 | 19.22 | 25.85 | −0.63 | 83.51 |
| MV = 15; PRCTILE = 95 | −6.18 | 10.18 | 4.04 | 17.42 | 23.18 | 3.08 | 12.40 | −9.82 | 0.78 | 30.75 | 31.05 | 6.16 | 123.04 |
| MV = 15; PRCTILE = 98 | −1.94 | 10.18 | 4.04 | 17.42 | 0.00 | −0.92 | 18.13 | −6.58 | 0.27 | 30.75 | 25.67 | 6.16 | 103.18 |
| MV = 20; PRCTILE = 90 | 8.62 | −0.27 | −0.39 | 19.01 | 25.34 | −3.99 | −4.03 | −3.76 | −11.80 | 19.35 | 16.89 | 0.43 | 65.39 |
| MV = 20; PRCTILE = 95 | 3.54 | 7.26 | 18.99 | 41.03 | 19.21 | −0.92 | 10.60 | −16.14 | −21.74 | 18.56 | 25.10 | 4.76 | 110.25 |
| MV = 20; PRCTILE = 98 | 2.69 | 13.57 | 2.61 | 43.15 | −1.83 | −4.60 | 20.25 | −7.21 | −22.25 | 28.73 | 25.10 | 6.16 | 106.38 |
| MV = 25; PRCTILE = 90 | 12.79 | −3.50 | 14.23 | 13.44 | 21.30 | 6.74 | −20.85 | −3.52 | −15.07 | 19.81 | 12.38 | −2.12 | 55.63 |
| MV = 25; PRCTILE = 95 | 8.94 | 9.23 | 7.07 | 30.87 | 17.87 | −3.47 | −9.95 | −21.56 | −17.47 | 33.01 | 20.57 | −0.63 | 74.49 |
| MV = 25; PRCTILE = 98 | 4.97 | 13.57 | −11.09 | 38.22 | −4.14 | −2.87 | 6.73 | −10.08 | −13.73 | 20.21 | 23.28 | 6.16 | 71.24 |
| Buy&Hold | −5.67 | −5.95 | −42.25 | 15.24 | 18.83 | −13.83 | 9.09 | 9.58 | 8.24 | 12.42 | 4.62 | 3.11 | 13.42 |

The table shows the P&L per year obtained by following either $I_t^{LTM}$ or the Buy and Hold strategy. Rows also indicate the sensitivity of the investment strategy to different values of the moving window (MV) and of the threshold of the empirical distribution (PRCTILE) used to identify phases of unstable market co-movements. Last column represents the P&L over the entire sample.

much less reactive as it can be seen; for instance, after the market rebound of the second half of 2009 and in the first part of 2016. Finally, lower performances with respect to our proposed indicator clearly emerge if we observe the P&L obtained by applying a strategy relying on the indicator introduced by ref. [49] (in brown) and the P&L derived from a strategy based only on the average correlation of stocks' returns (in cyan). All in all, this suggests that the behavioral features of market participants that we propose to capture through the use of the AC and correlation values are together instrumental for anticipating the dynamics of the underlying financial system.

We also report in Table 1 the annual P&L achieved by following either our proposed investment strategy or a simply Buy&Hold strategy (last row). We investigate the robustness of our findings to changes in the length of the moving window adopted to compute the empirical distribution of $I_t^{LTM}$ and in the threshold employed to define the extreme values for this indicator. We observe that our strategy over-performs the simple Buy&Hold strategy over the entire sample period (last column). Beside the fact that the proposed investment strategy does not produce positive P&L for all the periods and for all parameters configurations, results in Table 1 still support the predictive performance of our approach. While during market up-trends the signal produces P&L in lines with the Buy&Hold strategy, the timely identification of downward phases limits severe losses that, on the contrary, impact on the naive Buy&Hold strategy (see also Supplementary information, Section 3.6).

To quantitatively assess the performance of the proposed investment strategy, we employ a non-parametric approach. We proceed by first computing the true-positive, true-negative, false-positive, and false-negative calls of our investment strategy conditionally on some pre-determined percentiles of the distribution of the absolute values of the market returns. We consider returns larger than an $\alpha$ percentile, with $\alpha$ varying from 10% to 90%. Then, we compute the associated precision and

accuracy measures for each percentile (see Supplementary information, Section 3.8). From Fig. 3b, it clearly emerges that the capability of the proposed investment strategy in discriminating between positive and negative market movements increases as long as we select larger absolute values of market returns. This means that the trading strategy based on the LTM indicator correctly anticipates future changes in the aggregate stock price indices, especially around large market movement.

Finally, in the Section 3.6 of the Supplementary information, the P&L obtained from this strategy is compared against other investment alternatives such as strategies based on the DFA signals alone. We show that our approach outperforms the other strategies even when we do consider transaction costs. In fact, by assuming transaction costs of 10 basis-points for each portfolio rebalance, we still get a positive P&L of about 5.5% per year.

## Discussion
In a highly interconnected financial system, it is of paramount relevance to detect the emergence of an abrupt transition from a stable configuration to a state of instability[20,52–55]. It is against this background that our results are derived in the context of a financial market in which we investigate how the effects of linkages at the micro-level may bring changes to the macro-system[39]. The level of connectivity influences the probability of the system to remain stable. However, the role played by connectivity depends also on how the structure of the network interacts with additional factors which are specific for financial markets, such as investors heterogeneity, incentives to misbehave and price changes.

In this work, we have introduced an indicator that aims at detecting the emergence of instabilities in financial markets. The absence of a unified quantitative framework to properly formalize the laws of motion of financial markets motivates the use of instruments derived from the network theory to detect the emergence of discontinuities and their temporal evolution.

Changes in the market conditions are inspected through the analysis of the underlying system at different points in time. Phases of market instability are then assessed by the changes in the structure of the interactions among stock returns.

In particular, we have identified phases of accumulation of instability by detecting the emergence of a sub-graph of stocks characterized by both high cohesiveness among its members and long-range memory, which we relate to herding behaviors and positive feedbacks. We summarize the dynamic of this sub-graph through a synthetic indicator which we show to be able to detect temporary transitions of the underlying system. To test this approach, we have also proposed illustrative investment strategies to identify the emergence of up and down market phases according to the signals provided by the indicator. Our results show that this methodology can timely recognize phases of increasing instability that are likely to drive the underlying system into a new market configuration.

## Methods

**Detrended fluctuation analysis**. DFA[28–30] is employed in the first step of the analysis to filter stocks presenting long-term memory. The returns of these stocks are then clusterized according to their correlation values in order to identify the module approaching the phase transition towards a new equilibrium.

The DFA method comprises the following steps. In the first step, the data series $y(k)$, consisting in the stocks returns, is shifted by its mean $<y>$ and integrated (i.e., cumulatively summed) as follows:

$$x(k) = \sum_{i=1}^{k} [y(i) - <y>]. \tag{2}$$

In the second step, the transformed series is segmented in windows of various length $\Delta l$. For each segmentation, and repeatedly for all values of $\Delta l$, the summed data are fit with a polynomial $x_{\Delta l}(k)$. By this, the mean squared residual is found as:

$$F(\Delta l) = \sqrt{\frac{1}{L} \sum_{k=1}^{L} [x(k) - x_{\Delta l}(k)]^2}, \tag{3}$$

where $L$ is the total number of data points. In our analysis, we have applied a linear fit with $L$ set to 200 days. It is worth to remark that $F(\Delta l)$ can be viewed as the average of the summed squares of the residual computed in the windows.

Finally, a log–log graph of $F(\Delta l)$ against $\Delta l$ is drawn. This relationship is expected to be linear if power law scaling is present. In other words, a straight line on this log–log graph indicates statistical self-affinity expressed as $F(\Delta l) \propto (\Delta l)^{\alpha}$. The scaling exponent $\alpha$ is calculated as the slope of a straight line fit to the log–log graph of $\Delta l$ against $F(\Delta l)$ using least squares. This scaling parameter is a measure of the presence of self-similarity and, therefore, of long-term memory in the signal, as it tracks down the scaling of dispersion around a regressor for increasing window sizes. In particular, the value of $\alpha$ can describe the following signal behaviors: if $0 < \alpha < 0.5$, then the signal has long-term memory and it is anti-correlated; if $0.5 < \alpha < 1$, the signal has long-term memory and it is correlated; if $\alpha = 0.5$, the signal is uncorrelated (has no memory); finally, if $1 < \alpha < 2$, the signal is non-stationary.

The fluctuation function has a relationship with the AC of stationary process[56]. Indeed, the square of the fluctuation function $F(\Delta l)$ can be written as a function of the autocorrelation as:

$$F^2(\Delta l) = <y^2>(W(\Delta l) + \sum_{b=1}^{\Delta l-1} \mathrm{ACo}(b) L_b(\Delta l)) \tag{4}$$

being $\mathrm{ACo}(b)$ the autocorrelation function. Thus, in terms of AC, for a linear detrending, it is straightforward to compute $W(\Delta l)$ and $L_b(\Delta l)$, as:

$$W(\Delta l) = \frac{\Delta l^2 - 4}{15\Delta l}, \tag{5}$$

$$L_b(\Delta l) = \frac{1}{15(\Delta l^4 - \Delta l^2)}(-\Delta l + 15\Delta l^2 + 20\Delta l^3 + 15\Delta l4 + 25\Delta l^5). \tag{6}$$

The fluctuation function of DFA is therefore fully determined by the AC(1) and by the variance $V(1)$ of the process. For a long-range correlated process, these components are dominant on all time windows and hence a single scaling range with the correct exponent exists.

Our study starts by assessing the significance of the DFA coefficients considering the results computed from the original data and from surrogate data, namely, data obtained by independent time permutation for each stock returns. In other words, for a given time series we obtain its randomized (shuffled) counterpart by randomly rearranging time stamps attributed to each element in the series.

By comparing original results to those obtained for randomized data, we are able to wash out stocks that presents DFA coefficients in line with the one observed from the shuffled case. Basically, we identify the 5–95th percentiles of the DFA coefficients distribution as reference thresholds for assessing the statistical significance of the DFA value. Stocks with extreme values of the DFA, that is, stocks with DFA coefficients belonging to the tails of the distribution, will be then clusterized to obtain an indicator of price co-movement whose dynamics will be used to identify the occurrence of market instabilities and, accordingly, to distinguish between upward and downward market phases.

**The leading temporal module**. In what follows, we describe the methodology that allows us to identify a general signal indicating an imminent bifurcation. This signal is associated with the presence of an LTM, whose statistical properties reflect a transition of the underlying system to another state. In particular, it can be shown that, when a system is undergoing a bifurcation, the following general temporal and spatial properties hold: a group of stocks displays an average within PCC that drastically increases in absolute value; the average between PCC of stocks in this group and other stocks in the rest of the system will greatly decrease in absolute value; the average AC of stocks belonging to this group increases in absolute value. If all the three above-mentioned conditions are simultaneously satisfied, we call the group of stocks fulfilling these requirements the LTM of the system[51,57,58].

We now sketch the theoretical background at the basis of our indicator of market instability. Assume that the following discrete-time dynamical system describes the law of motion of a financial market, for example, in terms of stock prices or returns:

$$\mathbf{Z}(t + 1) = f(\mathbf{Z}(t); P) + \varepsilon(t), \tag{7}$$

where $\mathbf{Z}(t) = (z_1(t), ..., z_n(t))$ is a $n$-dimensional state vector representing stocks returns, $P = (p_1, ..., p_s)$ is an $s$-dimensional parameter vector representing slowly changing factors (e.g., news on earnings or profits, anticipated takeovers or mergers, etc.) and $\varepsilon = (\varepsilon_1, ..., \varepsilon_n)$ is a $n$-dimensional stochastic component with $\varepsilon_i$ Gaussian white noise with zero means and covariances $\kappa_{ij} = \mathrm{Cov}(\varepsilon_i, \varepsilon_j)$. In general, we assume $f : R^n \times R^s \to R^n$ is a nonlinear vector-valued function. In order to apply theoretical results on bifurcations of a general discrete-time dynamical model, we consider only the deterministic skeleton of the system, that is, we set $\varepsilon(t) = 0$. Furthermore, let us assume that the following conditions for Eq. (7) hold: $\bar{\mathbf{Z}}$ is a fixed point of (7), that is $\bar{\mathbf{Z}} = f(\bar{\mathbf{Z}}; P)$; there exists a value $P_c$ such that one or a complex conjugate pair of the eigenvalues of the Jacobian matrix of Eq. (7) evaluated at the fixed point $\bar{\mathbf{Z}}$ is equal to 1 in modulus; when $P \neq P_c$ the eigenvalues of the Jacobian matrix of (7) are generally not 1 in modulus.

These conditions, along with other transversality conditions, imply that the system undergoes a transition at $\bar{\mathbf{Z}}$ or a codimension-one bifurcation[59]. The parameter $P_c$, at which the transition for the equilibrium value $\bar{\mathbf{Z}}$ occurs, is called a bifurcation value (or a critical transition value) where a sudden qualitative or topological change takes place. The bifurcation is generic from a mathematical viewpoint, that is, almost all bifurcations for a general system satisfy these conditions. Around the fixed point $\bar{\mathbf{Z}}$, it is possible to linearize the system described by Eq. (7) as:

$$\mathbf{Z}(t + 1) \simeq \mathbf{J}(\mathbf{Z}(t) - \bar{\mathbf{Z}}), \tag{8}$$

where $\mathbf{J} = \mathbf{J}(P)$ denotes the Jacobian matrix of (7). By defining $\mathbf{X} = \mathbf{Z} - \bar{\mathbf{Z}}$, it is possible to shift the fixed point to the origin, and the system characterized by Eq. (8) can be re-written as:

$$\mathbf{X}(t + 1) = \mathbf{J}\mathbf{X}(t), \tag{9}$$

where $\mathbf{J}$ is a full-rank matrix that also depends on the vector $P$. Since the Jacobian matrix $\mathbf{J}$ is of full rank, then there exists a full-rank matrix $\mathbf{S}$ satisfying:

$$\mathbf{J} = \mathbf{S}\Lambda\mathbf{S}^{-1}. \tag{10}$$

By defining $\mathbf{Y} = \mathbf{S}^{-1}\mathbf{X}$, and reintroducing the stochastic component $\varepsilon$, the linearized version of the original system can be re-written as:

$$\mathbf{Y}(t + \Delta t) \simeq \Lambda\mathbf{Y}(t) + \varepsilon(t). \tag{11}$$

By fixing the value of parameter $P$ before reaching $P_c$, either $\mathbf{J}$ or $\Lambda$ is a constant matrix of full rank and we may end up with three cases: real and distinct eigenvalues, real and coincident eigenvalues, and complex eigenvalues.

If the sum of the dimensions of the eigenspaces with real eigenvalues is $n$, then there exists a non-singular matrix $\mathbf{S}$ satisfying $\Lambda = \mathbf{S}^{-1}\mathbf{J}\mathbf{S} = \mathrm{diag}(\lambda_1, ..., \lambda_n)$ being $\lambda_i$ the $i$th eigenvalue of the system (11). Without loss of generality, we may regard the first element $|\lambda_1|$ as being the nearest to 1, that is, the dominant eigenvalue, whose change leads to the state shift. If matrix $\mathbf{J}$ does not have linearly independent eigenvectors, there exists a non-singular matrix $\mathbf{S}$ making $\Lambda$ block diagonal. We can always move the block with the largest eigenvalue in modulus, which is also the nearest to 1, to the first entry of $\Lambda$. Finally, in the case of complex eigenvalues there is a non-singular matrix $\mathbf{S}$ making $\Lambda$ block diagonal where each two-dimensional block matrix has a pair of complex conjugated eigenvalues whose moduli are <1. As before we move the block in which the eigenvalues have the largest modulus to the first entry of $\Lambda$. Therefore, irrespective of which case occurs, the first element of $\Lambda$ is the dominant eigenvalue, that is, the one nearest to 1 in modulus, whose change

actually leads to the state shift from the fixed point. Furthermore, all the eigenvalues (or their moduli) of matrix $\Lambda$ are within $[0, 1)$ and there is at least one dominant eigenvalue approaching 1 in modulus when $P \to P_c$.

For simplicity, we shall show the statistical properties of the original variables $\mathbf{Z}$ considering only the case of real and distinct eigenvalues, but the same conclusion applies for the other two cases in a similar manner[58].

Since $\Lambda$ is a full diagonal matrix, we have the variance $V(\,\cdot\,)$, the covariance $C(\,\cdot\,)$, the auto-covariance $\mathrm{AC}(\,\cdot\,)$, and the Pearson correlation coefficient $\mathrm{PCC}(\,\cdot\,)$ of the autoregressive process expressed in Eq. (11) read as:

$$V\big(y_i(t)\big) = \frac{\kappa_{ii}}{1 - \lambda_i^2}, \tag{12}$$

$$C\big(y_i(t), y_j(t)\big) = \frac{\kappa_{ij}}{1 - \lambda_i \lambda_j}, \tag{13}$$

$$\mathrm{AC}\big(y_i(t), y_i(t-1)\big) = \frac{\lambda_i \kappa_{ii}}{1 - \lambda_i^2}, \tag{14}$$

$$\mathrm{PCC}\big(y_i(t), y_j(t)\big) = \frac{\kappa_{ij}}{\sqrt{\kappa_{ii}\kappa_{jj}}} \frac{\sqrt{\big(1 - \lambda_i^2\big)\big(1 - \lambda_j^2\big)}}{1 - \lambda_i \lambda_j}. \tag{15}$$

The dynamics of the original variable can be written as:

$$z_i(t) = s_{i1}y_1(t) + \cdots + s_{in}y_n(t) + \bar{z}_i, \; z_j(t) = s_{j1}y_1(t) + \cdots + s_{jn}y_n(t) + \bar{z}_j. \tag{16}$$

Thus, the variance and covariance of the original variables are given by:

$$\begin{aligned} V(z_i(t)) = {}& s_{i1}^2 V\big(y_1(t)\big) + \sum_{k=2}^{n} s_{ik}^2 V\big(y_k(t)\big) \\ & + \sum_{k,m=1,k\neq m}^{n} s_{ik}s_{im}\mathrm{PCC}\big(y_k(t), y_m(t)\big), \end{aligned} \tag{17}$$

$$\begin{aligned} C(z_i(t), z_j(t)) = {}& s_{i1}s_{j1}C\big(y_1(t)\big) + \cdots + s_{in}s_{jn}C\big(y_n(t)\big) \\ & + \sum_{k,m=1,k\neq m}^{n} s_{ik}s_{im}\mathrm{PCC}\big(y_k(t), y_m(t)\big). \end{aligned} \tag{18}$$

The correlation is given by:

$$\mathrm{PCC}(z_i(t), z_j(t)) = \frac{C(z_i(t), z_j(t))}{\sqrt{V(z_i(t))V(z_j(t))}}, \tag{19}$$

while the auto-covariance reads as:

$$\begin{aligned} \mathrm{AC}(z_i(t), z_i(t-1)) = {}& s_{i1}^2 \lambda_1 V\big(y_1(t)\big) + \sum_{k=2}^{n} s_{ik}^2 \lambda_k V\big(y_k(t)\big) \\ & + \sum_{k,m=1,k\neq m}^{n} s_{ik}s_{im}(\lambda_k + \lambda_m)\mathrm{PCC}\big(y_k(t), y_m(t)\big). \end{aligned} \tag{20}$$

Equations (19) and (20) relate the empirical signals of the original system (7) with the value assumed by the dominant eigenvalue of the latent system (11). It is worth to note that an increase of the variance, covariance, and autocorrelation of the original system could be due to both a proximity of a tipping point or a strong and unexpected exogenous shock in the stochastic component of the autoregressive process in (11).

The temporal and spatial statistical properties that signal an imminent bifurcation can thus be summarized as follows: if a variable $z_i$ is related to $y_1$, that is, $s_{i1} \neq 0$, then the absolute value of the auto-covariance $\mathrm{AC}(z_i(t), z_i(t-1))$ increases greatly as $\lambda_1 \to 1$; otherwise, it is bounded; if variables $z_i$ and $z_j$ are related to $y_1$, that is, $s_{i1} \neq 0$, $s_{j1} \neq 0$, then $|\mathrm{PCC}(z_i(t), z_j(t))| \to 1$ as $\lambda_1 \to 1$; if variables $z_i$ and $z_j$ are not related to $y_1$, that is, $s_{i1} = 0$, $s_{j1} = 0$, then $|\mathrm{PCC}(z_i(t), z_j(t))| \to a$ with $a \in (0, 1)$ as $\lambda_1 \to 1$; if only variable $z_i$ is related to $y_1$ but $z_j$ is not, that is, $s_{i1} \neq 0$, $s_{j1} = 0$, then $|\mathrm{PCC}(z_i(t), z_j(t))| \to 0$ as $\lambda_1 \to 1$.

**LTM identification**. Stocks $z_i$ in the system are represented as a dynamical temporal graph $G_t = (N_t, E_t)$ composed by $N_t$ nodes, while edges $E_t$ denote the pairwise correlation ($\mathrm{PCC}(z_i(t), z_j(t))$) between each pair of stocks' returns ($z_i(t), z_j(t)$) computed over a given moving window. This approach relies on the identification of two main sets of stocks: (i) the LTM denoted as $N_t^{\mathrm{LTM}}$ and (ii) the remaining stocks $N_t \backslash N_t^{\mathrm{LTM}}$ not belonging to the leading module. To detect whether the system is approaching a new equilibrium, we expect that[58,59]: (i) the absolute value of the auto-covariance of the time series of the LTM members in $N_t^{\mathrm{LTM}}$ increases; (ii) the absolute value of the correlation between stocks in the LTM increases as well; (iii) conversely, the absolute value of the correlation between a stock in $N_t^{\mathrm{LTM}}$ and another stock outside the LTM decreases to zero.

More practically, to identify the LTM we apply a hierarchical clustering procedure that distinguishes different groups or modules of stocks. We characterize

each identified module H by summarizing the statistical features reported above through a synthetic indicator. Let us denote the mean of the absolute value of the auto-covariance of the nodes in $N_t^{\mathrm{H}}$ as $<|\mathrm{AC}^{\mathrm{H}}|>$, the mean of the absolute value of the correlation coefficients between members of the H-th module as $<|\mathrm{PCC}_t^{\mathrm{H}}|>$, and let $<|\widetilde{\mathrm{PCC}_t^{\mathrm{H}}}|>$ be the analogous between stocks in $N_t^{\mathrm{H}}$ and the remaining stocks. The corresponding synthetic indicator for stocks within each module is defined accordingly as:

$$I_t^{\mathrm{H}} = \frac{<|\mathrm{AC}_t^{\mathrm{H}}|><|\mathrm{PCC}_t^{\mathrm{H}}|>}{<|\widetilde{\mathrm{PCC}_t^{\mathrm{H}}}|>}. \tag{21}$$

Then, the module with the highest value of $I_t^{\mathrm{H}}$ is assumed as the LTM of the underlying system and the corresponding indicator, labeled $I_t^{\mathrm{LTM}}$, is employed for monitoring the reinforcement of market instabilities. This indicator is expected to sharply increase when a new phase is about to be reached by the underlying system, representing therefore an effective marker for the identification of a cumulative process leading to a new system configuration[51,58,60]. Hence, we expect the LTM to emerge more clearly when the system is experiencing a transition, meaning that its members become more cohesive and distinct from the rest of the network. In Supplementary information we present a pseudo-code that formalizes the procedure (see Supplementary Fig. 6).

**Reporting summary**. Further information on research design is available in the Nature Research Reporting Summary linked to this article.

## Data availability
The data that support the findings of this study are available on request from the corresponding author A.S. The data are not publicly available due to privacy restrictions.

## Code availability
Codes are available upon request.

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

## Acknowledgements

We acknowledge funding from the National Research Project (PNR) CRISIS Lab. We thank anonymous Reviewers and the Editor for their detailed and constructive feedback on our paper.

## Author contributions

A.S., A.F., N.P., F.P., and S.B.: designed the research, performed the research, analyzed the data, and wrote the paper.

## Competing interests

The authors declare no competing interests.
