## [Peer Review File · Nature Communications]

Reviewers' comments:

Reviewer #2 (Remarks to the Author):

This paper proposes an indicator based on the covariance of stock returns, which 'aims to detect the emergence of instabilities in financial markets' and makes claim about the value of this indicator based on (in-sample) performance on daily stock returns over 10 years.

The indicator, which is based on Chen et al (2012), may be described in fact simply in terms of a clustering algorithm for stocks based on a moving- window correlation analysis followed by the computation of a ratio involving absolute auto-covariances and pairwise correlations (Equation 1). The formula for the indicator is ad-hoc and not justified in terms of any model of theoretical analysis. The description of the indicator is obscured by a lengthy detour in Section 1 with needless jargon ('Leading Temporal Module', 'topological analysis' etc).

The main justification for the indicator is based on plots shown in Section 3. The authors claim that the LTM indicator 'starts to increase prior to the 2008 global financial crisis and reaches its peak approximately in correspondence of the outbreak of the capital markets'. Plot D shows in fact that, like other indicators based on correlation analysis of returns, LTM peaks *after* not before, the Lehman default, so the above claim is misleading. If there is any claim that LTM is an 'early warning indicator' then this should be borne out with a careful lead-lag analysis accounting for estimation errors, not with a couple of plots.

A statistical methodology is missing and details on estimation error are not given. Using daily data with an averaging window of 1 year (?), one has 250 points, but sample estimates seem to be treated as exact values without any analysis of statistical error, which is surprising given the small sample sized used.

Given the potential to detect spurious 'early warning' signals, it would have been useful to test the methodology on simulated or surrogate data. How is this indicator more useful than, say, an average correlation measure?

The title and the text make frequent reference to 'graph theoretical tools' and 'topological analysis' but in fact no 'graph theoretical tools' or 'topological analysis' are found in the paper (unless one considers data clustering to be 'topological analysis').

Finally, the paper fails to make any link with the (vast) literature on network models in finance which, in fact, does make use of graph theoretical tools to identify stability conditions in financial networks.

Given

- the lack of clear methodology,
- the lack of interpretability of the proposed indicator and
- the presence of unsubstantiated claims regarding its empirical performance

we do not find this paper suitable for publication in NATURE COMMUNICATIONS.

We recommend the authors to rewrite the paper with more detail on the methodology, to substantiate their claim of the 'early warning' nature of the LTM indicator with a more careful lead-lag analysis, to perform at least some simulation tests or tests with surrogate data as a benchmark. If this is done, I would recommend them to send it to a financial econometrics journal where readers and reviewers will be better able to judge its novelty.

Reviewer #3 (Remarks to the Author):

The paper puts forward an idea to use changes in the topological structure of return linkages between financial assets (stocks) to detect changes on the financial market. To do that, the paper presents a new topological indicator calculated from financial returns data. An empirical application is performed where the new indicator is shown to predict market changes better than benchmark models.

The general idea is not entirely new, but as far as I can tell, the specific approach is new in the context of financial markets and on a more subjective note, it is interesting. The indicator presented in the paper (Eq. 1 & Eq. 15 in SI) is based on identifying a subset of vertices (stocks), called Leading Temporal Module (LTM) in a graph that are:

- i) very close to each other (highly interconnected returns),
- ii) very far from the rest of the stocks (less interconnected returns) and that also
- iii) show higher (absolute value) auto-covariance (of returns) with each other.

The third criteria seems to be different from what was used before in Chen et al., (2012), where standard deviation is used, i.e. basically auto-covariance at order 0. Using historical data, such group of stocks is identified for each time (trading day). The newly proposed indicator is calculated for each day and the indicator gives higher values for higher correlation of these stocks with each other, higher auto-covariance between these stocks and lower correlation with the rest of the stocks. Higher values of the indicator indicate changes in the structure of market linkages.

I believe that the approach can be used (adapted) when describing complex relationships of any systems that evolve over time, e.g. relationships between people (social networks),...

I think that the overall approach presented in the paper is interesting and that it has the potential to influence the way we think about the practical purpose of complex networks. It certainly made me realize several new ideas.

I have several comments, which I hope could improve the current version of the paper.

1) I advise to re-think the way how the introduction of the paper is written. Either the main message is the story about detecting changes in the market, by the means of a new indicator, or the main message is about presenting new indicator and giving an empirical application. After reading the current version of the Introduction I was not sure which one is it. Only latter it seems that the first one.

2) What I missed is a connection to other topological indicators that might be related to the one presented in the paper (community detection algorithms).

3) In the abstract the sentence: "Our study reveals the emergence of a sub-graph of stocks, among those referring to a benchmark financial index, as the marker of an out-of-equilibrium transition of the underlying system." is hard to understand in the given context and perhaps instead of the term 'financial index' the term 'market index' is more useful.

4) What tends to be the industry or market capitalization composition of the LTM stocks? Any pattern there?

5) It might help for the general reader if you would explain, intuitively following terms that are used in the Introduction section, 'endogenous instabilities', 'positive feedbacks', 'herding behavior'.

6) I found the following sentence unnecessary: 'Interestingly, if expectations about the future price of a certain asset are high, then the demand for that asset is likely to increase and the realized price will also be high, so that a large cumulative move may ensue, leading to the broken balance between autonomous conducts and peer influence Preis et al. (2011).'

7) In Section 2 you refer to Chen et al., (2012) where you write about the 'system approaching a tipping point', in Chen et al., (2012) their indicator used standard deviation of group components and in your paper absolute auto-covariances are used. Could you elaborate in the paper in what way is your indicator different from that of Chen et al., (2012)? What is the intuitive motive of using these indicators? The technical details and derivations are shown in Chen et al., (2012) and in your SI, but in the text an intuitive explanation would be helpful.

- 8) You use notation N_t^0 to denote stocks that are not in $N_t^{\{LTM\}}$ but latter you seem to use N_t^0 to denote stocks that are not in N_t^H . It does not disturb me, but if I am correct, your notation here is not precise.
 - 9) The idea that there is a group that becomes more cohesive and distinct from the rest of the network before the system is experiencing a transition is nice. And hopefully useful. I think that you could elaborate on why the approach should work in the first place. Perhaps, when market is driven by changes in one or two sectors, returns from stocks from these sectors get more correlated, more distinct and perhaps persistent. But why only one such group? Have you considered a case with multiple groups?
 - 10) The use of DFA pre-filtering for market data seems to be slightly odd, but it might be my background, where I would expect an econometric approach (ARMA-GARCH filtering). This is just an observation of mine.
 - 11) I recommend to name the axis in Panel A, B of the Figure 1.
 - 12) I would advise to re-phrase the following sentence: 'while correlation values of the LTM members indicate the presence of a bunch of stocks that highly co-move among themselves and...'
 - 13) 'Changes in the LTM composition is important' – are.
 - 14) In Figure 2, LTM and Stoxx should have a slightly different color or line style.
 - 15) '...the most recent value of the indicator is compared against its empirical distribution computed over the previous working month.' Against previous calendar month or just previous approx. 22 days that correspond to number of trading days in a month. The later seems to be a more usual choice.
 - 16) 'To detect the direction of the market trend, we exploit information contained in the average return of the LTM members.' Over which period? Also over the past month?
 - 17) You report P&L of 60% please annualized this measure.
 - 18) In the SI you report results from other benchmarks, I think they should be in the main paper if possible. At least, the Buy & Hold. Also you might consider some other model based benchmark strategy that predicts market up and down turns.
- Thank you for the interesting paper.

Reviewer 1

We are grateful for the Reviewer's comments and suggestions, which we used to improve the paper. We reproduce the comments below and reply to the points raised by the Reviewer:

1. *This paper proposes an indicator based on the covariance of stock returns, which 'aims to detect the emergence of instabilities in financial markets' and makes claim about the value of this indicator based on (in-sample) performance on daily stock returns over 10 years. The indicator, which is based on Chen et al (2012), may be described in fact simply in terms of a clustering algorithm for stocks based on a moving- window correlation analysis followed by the computation of a ratio involving absolute auto-covariances and pairwise correlations (Equation 1). The formula for the indicator is ad-hoc and not justified in terms of any model of theoretical analysis. The description of the indicator is obscured by a lengthy detour in Section 1 with needless jargon ('Leading Temporal Module', 'topological analysis' etc).*

We thank the Reviewer for these observations that help us in improving the overall quality of the paper. As the Reviewer correctly states, our proposed measure is based on the work of Chen et al. (2012) who proposed Dynamical Network Biomarkers (DNBs) to detect EWSs for the progression of complex diseases using throughput data. Differently from Chen et al. (2012), who look at the dispersion of the time series near a critical transition, in the detection of EWSs of financial crisis, beside variables co-movement, we are also keen in quantifying the self-similarity of the returns through the inclusion of autocovariance values. In this way we are able to capture two behavioral features at the ground of many financial crisis, namely herding behaviors and positive feedbacks.

The formula for the indicator that we propose to study the underlying market is grounded on a precise theoretical analysis which is reported in the Supplementary Information of the paper, where we describe the analytical background that leads to the definition of the indicator for the identification of the Leading Temporal Module. Specifically, we start by observing that although the behavior of financial markets is very complicated, in principle it could be described by state equations in a high-dimensional space with a large number of variables and parameters. Unfortunately, this task is practically not feasible and, therefore, it is necessary to employ aggregation methods to address the dynamics of a system. Aggregation of variables has been explicitly used in economics to study and to evaluate the dynamics of systems of great size and complexity (see Simon and Ando (1961)). This behavior has led to the practice of employing low dimensional systems to describe the evolution of high-dimensional aggregate environments (see e.g Barunik and Kukacka (2015); Diks et al. (2015); Diks and Wang (2016)). Starting from this common approach, we employ the dynamical system theory to show that a high dimensional system can be expressed in a very simple form but in an abstract or latent phase space and that, by detecting some particular empirical signals, we can infer its critical transition. In other words, despite the fact that the system variables are generally unobservable in the abstract phase space, however, if some of these variables first cross the transition point, then a group of state and observable variables will display some particular statistical features (see also Chen et al. (2012); Fushing et al. (2014)). This group of

variables corresponds to the observable LTM in the original state space. In particular, in the Supplementary Information, we have shown the conditions to identify the LTM in the original state space, relating its statistical properties to the patterns of the unobservable variables in the abstract phase space.

Moreover, we have employed a Lotka-Volterra model of stock dynamics that serves as an example to demonstrate that the generated time series present statistical properties related to the patterns of the unobservable variables in the abstract phase space. According to the Reviewer suggestion we have enlarged the analysis on the simulated series of the Lotka-Volterra model by adding the following paragraph in the Supplementary Information:

“To mimic the possible system dynamics far and near a transition point, we simulate the system with different parameter values and then we compute the statistical measures which constitute the indicator. Figure 1 represents a situation in which the system is far from a transition point while in Figure 2 the system is near a transition phase. In the top left panels of Figures 1 and 2 in particular, we display the resulting dynamic of the four modeled stock prices. Specifically, while far from transition, the time series exhibit small fluctuations and a relatively low autocovariance (see top right panel of Figure 1), close to the tipping point the fluctuations exhibit higher volatility and autocovariance (top right panel of Figure 2). Also the correlation between the stocks becomes stronger (lower panels of Figures 1 and 2).“

Finally, we have also added to the Supplementary Information a study to test the methodology on simulated or surrogate data in order to provide further validation of our technique. Details on this new part are given below while answering to point three.

As the Reviewer suggested, we have also deeply modified the introduction of the paper emphasizing the main message, which relates to the detection of changes in the market by means of a new indicator for which we have better described its properties and economic meaning. In addition, we have removed the references to the topological analysis to avoid misunderstandings.

Figure 1: **Lotka-Volterra model of financial relationships far from phase transition:** model simulations illustrating generic indicators far from the tipping point. Parameter values are: $b_1 = b_2 = 1$, $b_3 = b_4 = 0.5$, $\sigma = 0.15$ and $\mu = 1$.

Figure 2: **Lotka-Volterra model of financial relationships near phase transition:** model simulations illustrating generic indicators near the tipping point. Parameter values are: $b_1 = b_2 = 1$, $b_3 = b_4 = 1.1$, $\sigma = 0.15$ and $\mu = 1$.

2. *The main justification for the indicator is based on plots shown in Section 3. The authors claim that the LTM indicator “starts to increase prior to the 2008 global financial crisis and reaches its peak approximately in correspondence of the outbreak of the capital markets”. Plot D shows in fact that, like other indicators based on correlation analysis of returns, LTM peaks after not before, the Lehman default, so the above claim is misleading. If there is any claim that LTM is an ‘early warning indicator’ then this should be borne out with a careful lead-lag analysis accounting for estimation errors, not with a couple of plots.*

We thank the Reviewer for this observation which helps us in clarifying the functioning of the proposed indicator. The dynamics of the LTM sub-graph aims at identifying market phases characterized by the strengthening of price co-movements responsible for the transition of the whole underlying market away from its current configuration. In this regards, we are simply referring to the detection of a cumulative process of market instability that leads to a tipping point. It is worth noting that we refer to the dynamics of the LTM indicator with respect to its historical pattern, thus a significant deviation of the indicator from its past values is signaling a probable and relevant variation in the current configuration of the system. Instead, we propose a simple yet intuitive investment approach to relate the information provided by the LTM and the direction of the underlying market by means of the returns of the LTM members.

Hence, the Reviewer is right when states that the indicator peaks, for instance, after, and not before, the Lehman default. Nevertheless, it is true that the LTM indicator starts to increase prior to the 2008 global financial crisis and it mostly and rapidly grows in correspondence of the outbreak of the capital markets when it reaches a local maximum. After that, it peaks when the financial conditions are improving, thus enlightening another market phase. That peak at the beginning of 2009 thus indicates that the system is undergoing a transition from a negative equilibrium (due to the financial crisis) to a positive one (due to the rebound of the economy after the crisis). According to these observations, we have modified the aforementioned sentence in the manuscript as: “In a nutshell, I_t^{LTM} increases and assumes higher values around periods of market instability than during a tranquil period. For instance, during the 2008 global financial crisis, I_t^{LTM} starts to increase prior to the outbreak of the market and it reaches a local maximum approximately in correspondence of the onset of the crisis.”.

The Reviewer also suggests to perform a lead-lag analysis to investigate the early warning properties of the proposed indicator. Despite this approach could be useful for evaluating the anticipatory properties of the indicator, our proposed indicator only indicates that the system is approaching a bifurcation, without suggesting the direction in which the system will jump (namely, a transition from an economically good equilibrium to a bad one, or vice-versa). To cope with the above issue, we rely on a non-parametric analysis investigating the properties of the investment strategy based on the dynamics of the I^{LTM} and on the average returns of the stocks composing the LTM. Namely, we measure the amount of times our investment strategy is able to correctly predict the one-step ahead market direction of the underlying index. This computation is repeated for different percentiles of the returns distribution of the reference market index and it allows us to investigate the capability of the investment strategy to correctly identify market movements of different scales. In the Supplementary Information

we have included the results of this analysis, which are expressed in the following paragraphs:

“To further quantify the ability of the proposed investment strategy to correctly identify different market phases, we employ standard approaches derived from non-parametric analysis. We attempt at measuring the performance of the investment strategy in discriminating between up and down market movements, for different thresholds of the distribution of its absolute returns. In other words, we count how many times the direction of the reference market price is correctly identified conditionally on the size of the absolute values of the returns. That is, how many times the strategy correctly predicts the direction of the absolute returns of the reference market index that are larger than some fixed percentiles of the distribution? Answering this question will shed light on the functioning of the proposed indicator and on the capability of the resulting investment strategy to forecast, at least, large market movements.

We proceed by first computing the true positive (TP), true negative (TN), false positive (FP) and false negative (FN) calls of our investment strategy for the absolute returns of the market index that are greater than some pre-determined percentiles of their distribution. We let the percentiles vary from 10% to 90% and, for each percentile, we compute the precision and accuracy measures as:

$$Accuracy = (TP + TN)/(TP + TN + FP + FN) \quad (1)$$

$$Precision = (TP)/(TP + FP) \quad (2)$$

The accuracy provides information on how close a value is to its true value while the precision refers to how repeatable a measurement is.

From Figures 3 and 4 it clearly emerges that the capability of the proposed investment strategy in discriminating between positive and negative returns increases as long as the returns have a larger magnitude. This reinforces the idea that the trading strategy based on the LTM indicator correctly anticipates at least some future changes in the aggregate stock price indices, especially around large market movements.

In the main text we have also reported the analysis related to the case of the STOXX Asia/Pacific 600.

Figure 3: **Accuracy and precision for the STOXX Asia/Pacific 600.** The figure reports the accuracy and precision measures of the proposed investment strategy conditionally on the forecast of the market index returns that are greater than a certain percentile of their distribution in absolute terms.

Figure 4: **Accuracy and precision for the STOXX North America 600.** The figure reports the accuracy and precision measures of the proposed investment strategy conditionally on the forecast of the market index returns that are greater than a certain percentile of their distribution in absolute terms.

3. *A statistical methodology is missing and details on estimation error are not given. Using daily data with an averaging window of 1 year (?), one has 250 points, but sample estimates seem to be treated as exact values without any analysis of statistical error, which is surprising given the small sample sized used. Given the potential to detect spurious 'early warning' signals, it would have been useful to test the methodology on simulated or surrogate data. How is this indicator more useful than, say, an average correlation measure?*

The Reviewer has risen a point that allows us to improve the quality of the paper by inserting statistical details of our proposed indicator. In particular, for assessing the statistical variability of our indicator we have computed error bounds by performing bootstrapping that allows us to assign a measure of accuracy to our indicator (see Figure 1, panel D). Basically, We have estimated the variability of our indicator by re-sampling 500 time the stocks' returns from the empirical distribution of the observed data, and computing for each run the LTM indicator. We then find the 5-95% confidence intervals to establish the uncertainty of the indicator.

Moreover, as suggested by the Reviewer we have also tested the methodology on simulated data. We have used a reverse engineering process by assuming that the data generating process takes the form expressed in Equations (5) and (10) of the Supplementary Information. Then we have simulated different configurations of the Jacobian matrix of the system, which are assumed as reference points for the system behavior near and far a bifurcation point. Finally we have computed the statistics of the LTM components in the two cases (far and near the phase transition) to find a connection between the generated time series and the behavior of the underlying system. We have added the following paragraphs into the Supplementary Material of the paper:

“The emergence of a group of stocks whose behavior signals that the underlying system is undergoing a transition phase can be related to the degree of decomposability¹ of the Jacobian matrix (see Simon and Ando (1961); Ando and Fisher (1963); Simon (1996); Courtois (2014)) of such system. The decomposability of the Jacobian matrix \mathbf{J} in Eq. (5) is indeed intimately linked to the appearance of a transition phase and therefore related to the empirical signals leading to the emergence of the LTM.

To analyze the behavior of the indicator I^{LTM} as long as \mathbf{S} and $\mathbf{\Lambda}$ vary, we generate synthetic data from the following data generating process that reproduces the model described by Eqs. (5) and (10):

$$\begin{aligned}\mathbf{Z}(t) &= \mathbf{S}\mathbf{Y}(t) \\ \mathbf{Y}(t+1) &= \mathbf{\Lambda}\mathbf{Y}(t) + \varepsilon(t)\end{aligned}$$

We consider different configurations of \mathbf{S} and $\mathbf{\Lambda}$ by changing the features of the Jacobian matrix \mathbf{J} and then by computing its eigenvalues $\mathbf{\Lambda}$ and eigenvectors \mathbf{S} . The Jacobian matrix takes the following form:

$$\mathbf{J} = \mathbf{J}^* + \delta\hat{\mathbf{J}}$$

¹A decomposable matrix is a square matrix such that a rearrangement of rows and columns leaves a set of square submatrices on the principal diagonal and zeros everywhere else.

where \mathbf{J}^* is diagonal and $\hat{\mathbf{J}}$ encompasses the values outside the main diagonal. In this way, tuning the parameter δ , we can let the degree of decomposability of \mathbf{J} assume different levels. We study how the indicator reacts to these changes. For this purpose we have produced two series of synthetic datasets both having $n = 20$ stocks and whose price pattern has been simulated for $t = 250$ periods. Moreover, we have set the variance of ε to 0.2 and, for the first dataset we have assumed $\delta = 10^{-5}$, while in the second case $\delta = 10^{-1}$. The first value of δ implies a quasi-diagonal matrix, thus a system far from the transition, in which each stock is governed by its own dynamic. The second value of δ , on the other hand, populates the off-diagonal entries of \mathbf{J} , whose elements are higher with respect to the first case, and thus implying that the variables are no longer independent and that diffusion processes, like herding behaviors, dominate the dynamics of the system. This second case postulates a system near the transition phase.

Figure 5 reports the histograms with the values of I^{LTM} generated from the different simulations. In Figure 5 we can better observe how the values of I^{LTM} increase as long as δ increases, thus as long as the system approaches a critical point. The distribution of I^{LTM} becomes more skewed and the average value of I^{LTM} , depicted by the dashed red line, grows. Finally, in Figure 6 we also report the histograms for the components of I^{LTM} derived from the different simulations obtained far and near the transition point. The figure shows that near the transition point there exists an increasing cohesiveness among stocks in each module and also an increasing autocovariance.”

Figure 5: **Histograms of I^{LTM} for different values of δ :** On the top, we report the distribution of I^{LTM} related to $\delta = 10^{-5}$ while the bottom panel shows the distribution of I^{LTM} for $\delta = 10^{-1}$. Near the transition phase, the distribution of I^{LTM} becomes more skewed and also the average value of I^{LTM} , depicted by the dashed red line, increases.

Figure 6: **Histogram of I^{LMT} components for different values of δ** : the upper panels refer to the I^{LMT} components distribution, namely the within PCC, the between PCC and the autocovariance related to $\delta = 10^{-5}$, while the lower panel show the distribution of the I^{LMT} components for $\delta = 10^{-1}$. Near the transition point, while the within module correlation and autocovariance increases, simulation shows that the between correlation remains approximately at the same magnitude.

4. *The title and the text make frequent reference to 'graph theoretical tools' and 'topological analysis' but in fact no 'graph theoretical tools' or 'topological analysis' are found in the paper (unless one considers data clustering to be 'topological analysis').*

According to the Reviewer's point, in the revised manuscript we have removed the references to graph theoretical tools and topological analysis since these terms may be misleading.

5. *Finally, the paper fails to make any link with the (vast) literature on network models in finance which, in fact, does make use of graph theoretical tools to identify stability conditions in financial networks.*

We agree with the Reviewer's comment which underlines a lack in connecting our paper with the existing literature on the broad field of financial networks. The first version of our manuscript mostly related our approach to other analytical techniques that aim at studying how cross-market linkages, comovements and interdependencies between assets contribute to explain the sustainability of financial markets, with a particular focus on the role of herding behaviors and positive feedbacks. Nonetheless, in the revised manuscript, we receive the Reviewer's point and we better link the present work with the existing literature on financial networks and stability conditions in financial networks. In fact, in recent years the fields of financial networks models and their applications to financial stability have attracted a growing interest both among scholars and practitioners. This research line has built both on the literature on complex networks (e.g., Caldarelli (2007); Barrat et al. (2008); Battiston et al. (2010); Barabási et al. (2016)) and on the literature in economic networks (e.g., Jackson and Watts (2002); Goyal and Vega-Redondo (2005)). Since many interactions that take place in financial markets can be represented as a network of financial linkages between institutions, a significant fraction of research in systemic risk has been devoted to the study of financial networks (see e.g., Haldane and May (2011); Fouque and Langsam (2013)). An advantage of modeling financial systems as complex networks is that we can directly analyze complex feedback between micro and macro phenomena, where the structure of the networks plays an essential role in leading micro events to macro behaviors. Moreover, our approach belongs to the stream of literature that focuses on network models of financial markets that has been developed outside traditional economics. In fact, in the presence of imperfect information and bounded rationality, there is more room for externalities that players are not able to avoid and systematic mispricing of classes of assets. All these elements build up together and may lead to systemic risk and eventually crisis. In this respect, the discipline of financial networks turns out to be relevant, as it allows to better understand the emergence of systemic effects and design tools and policies to stabilize and mitigate them (see e.g., Nier et al. (2007); Acemoglu et al. (2015); Bardoscia et al. (2017); Battiston and Martinez-Jaramillo (2018)). In general, we can state that the level of connectivity influences the probability of the system to remain stable. However, the role played by connectivity depends also on how the structure of the network interacts with additional factors which are specific for financial markets, such as investors and banks heterogeneity, incentives to misbehave and price changes. Stylized graph-theoretical models, as the ones introduced by Allen and Gale (2000); Freixas et al. (2000); Leitner (2005); Babus (2005, 2016), all agree that graph incompleteness increases systemic

risk and instability. More complex analytical and numerical models show how connectivity influences stability. In particular, connectivity and size of shocks interact in determining how a financial system may move from a state of stability to a state of instability in an abrupt way (see Gai and Kapadia (2010); Gai et al. (2011); Amini et al. (2012); Acemoglu et al. (2013)). It is against this background that our results are derived in the context of a financial market in which we investigate how the effects of linkages at the micro-level may bring changes on the macro-system (see also Corsi et al. (2016) for a similar approach). Overall, we can state that these aspects rise relevant scientific challenges and generate prominent policy and societal implications. Due to the limitations in the number of pages, we have decided to report only a brief part of the above discussion in the revised manuscript, as well as the corresponding references, and leave the entire discussion in the present letter which, we hope, will help to fit our work into the burgeoning literature on financial networks.

References

- Acemoglu, D., A. Ozdaglar, and A. Tahbaz-Salehi, 2013: The network origins of large economic downturns. Tech. rep., National Bureau of Economic Research.
- Acemoglu, D., A. Ozdaglar, and A. Tahbaz-Salehi, 2015: Systemic risk and stability in financial networks. *American Economic Review*, **105** (2), 564–608.
- Allen, F., and D. Gale, 2000: Financial contagion. *Journal of Political Economy*, **108** (1), 1–33.
- Amini, H., R. Cont, and A. Minca, 2012: Stress testing the resilience of financial networks. *International Journal of Theoretical and Applied Finance*, **15** (01), 1250 006.
- Ando, A., and F. M. Fisher, 1963: Near-decomposability, partition and aggregation, and the relevance of stability discussions. *International Economic Review*, **4** (1), 53–67.
- Babus, A., 2005: Contagion risk in financial networks. *Financial Development, Integration and Stability*, Edward Elgar, 423–440.
- Babus, A., 2016: The formation of financial networks. *The RAND Journal of Economics*, **47** (2), 239–272.
- Barabási, A.-L., and Coauthors, 2016: *Network Science*. Cambridge university press.
- Bardoscia, M., S. Battiston, F. Caccioli, and G. Caldarelli, 2017: Pathways towards instability in financial networks. *Nature Communications*, **8**, 14416.
- Barrat, A., M. Barthelemy, and A. Vespignani, 2008: *Dynamical processes on complex networks*. Cambridge university press.
- Barunik, J., and J. Kukacka, 2015: Realizing stock market crashes: stochastic cusp catastrophe model of returns under time-varying volatility. *Quantitative Finance*, **15** (6), 959–973.

- Battiston, S., J. B. Glattfelder, D. Garlaschelli, F. Lillo, and G. Caldarelli, 2010: The structure of financial networks. *Network Science*, Springer, 131–163.
- Battiston, S., and S. Martinez-Jaramillo, 2018: Financial networks and stress testing: Challenges and new research avenues for systemic risk analysis and financial stability implications. Elsevier.
- Caldarelli, G., 2007: *Large scale structure and dynamics of complex networks: from information technology to finance and natural science*, Vol. 2. World Scientific.
- Chen, L., R. Liu, Z.-P. Liu, M. Li, and K. Aihara, 2012: Detecting early-warning signals for sudden deterioration of complex diseases by dynamical network biomarkers. *Scientific Reports*, **2**, 342.
- Corsi, F., S. Marmi, and F. Lillo, 2016: When micro prudence increases macro risk: The destabilizing effects of financial innovation, leverage, and diversification. *Operations Research*, **64** (5), 1073–1088.
- Courtois, P. J., 2014: *Decomposability: queueing and computer system applications*. Academic Press.
- Diks, C., C. Hommes, and J. Wang, 2015: Critical slowing down as an early warning signal for financial crises? *Empirical Economics*, 1–28.
- Diks, C., and J. Wang, 2016: Can a stochastic cusp catastrophe model explain housing market crashes? *Journal of Economic Dynamics and Control*, **69**, 68–88.
- Fouque, J.-P., and J. A. Langsam, 2013: *Handbook on systemic risk*. Cambridge University Press.
- Freixas, X., B. M. Parigi, and J.-C. Rochet, 2000: Systemic risk, interbank relations, and liquidity provision by the central bank. *Journal of Money, Credit & Banking*, **32** (3), 611.
- Fushing, H., Ò. Jordà, B. Beisner, and B. McCowan, 2014: Computing systemic risk using multiple behavioral and keystone networks: The emergence of a crisis in primate societies and banks. *International Journal of Forecasting*, **30** (3), 797–806.
- Gai, P., A. Haldane, and S. Kapadia, 2011: Complexity, concentration and contagion. *Journal of Monetary Economics*, **58** (5), 453–470.
- Gai, P., and S. Kapadia, 2010: Contagion in financial networks. *Proceedings of the Royal Society A: Mathematical, Physical and Engineering Sciences*, **466** (2120), 2401–2423.
- Goyal, S., and F. Vega-Redondo, 2005: Network formation and social coordination. *Games and Economic Behavior*, **50** (2), 178–207.
- Haldane, A. G., and R. M. May, 2011: Systemic risk in banking ecosystems. *Nature*, **469** (7330), 351.
- Jackson, M. O., and A. Watts, 2002: The evolution of social and economic networks. *Journal of Economic Theory*, **106** (2), 265–295.
- Leitner, Y., 2005: Financial networks: Contagion, commitment, and private sector bailouts. *The Journal of Finance*, **60** (6), 2925–2953.

- Nier, E., J. Yang, T. Yorulmazer, and A. Alentorn, 2007: Network models and financial stability. *Journal of Economic Dynamics and Control*, **31 (6)**, 2033–2060.
- Simon, H. A., 1996: *The architecture of complexity*. Cambridge, MA: MIT Press.
- Simon, H. A., and A. Ando, 1961: Aggregation of variables in dynamic systems. *Econometrica*, **29 (2)**, 111–138.

Reviewer 2

The paper puts forward an idea to use changes in the topological structure of return linkages between financial assets (stocks) to detect changes on the financial market. To do that, the paper presents a new topological indicator calculated from financial returns data. An empirical application is performed where the new indicator is shown to predict market changes better than benchmark models. The general idea is not entirely new, but as far as I can tell, the specific approach is new in the context of financial markets and on a more subjective note, it is interesting. The indicator presented in the paper (Eq. 1 & Eq. 15 in SI) is based on identifying a subset of vertices (stocks), called Leading Temporal Module (LTM) in a graph that are: i) very close to each other (highly interconnected returns), ii) very far from the rest of the stocks (less interconnected returns) and that also iii) show higher (absolute value) auto-covariance (of returns) with each other. The third criteria seems to be different from what was used before in Chen et al., (2012), where standard deviation is used, i.e. basically auto-covariance at order 0. Using historical data, such group of stocks is identified for each time (trading day). The newly proposed indicator is calculated for each day and the indicator gives higher values for higher correlation of these stocks with each other, higher auto-covariance between these stocks and lower correlation with the rest of the stocks. Higher values of the indicator indicate changes in the structure of market linkages. I believe that the approach can be used (adapted) when describing complex relationships of any systems that evolve over time, e.g. relationships between people (social networks). I think that the overall approach presented in the paper is interesting and that it has the potential to influence the way we think about the practical purpose of complex networks. It certainly made me realize several new ideas. I have several comments, which I hope could improve the current version of the paper.

1. *I advise to re-think the way how the introduction of the paper is written. Either the main message is the story about detecting changes in the market, by the means of a new indicator, or the main message is about presenting new indicator and giving an empirical application. After reading the current version of the Introduction I was not sure which one is it. Only latter it seems that the first one.*

We are grateful to the Reviewer for her/his observation which actually allowed us to rethink about the main message of our work and how to better state it in the Introduction. The main goal of the paper is, in fact, that of detecting changes in market phases as a possible signal of the occurrence of crises; and this is done by the definition of an indicator which exploits the information embedded in the correlation matrices related to a market index.

Our methodology identifies a group of stocks, the Leading Temporal Module (LTM), whose statistical properties reflect the transition of the market into a crisis state. We define an indicator of the emergence of market discontinuities based on the autocovariance of the stocks in the LTM and on the ratio between the stocks' correlations within this group and the correlations between these stocks and those outside the leading module. We rely on the temporal and spatial properties of the LTM to build an aggregate and flexible indicator,

which we employ to detect the occurrence of significant changes in financial markets. In particular, we aim at predicting distress phases on aggregate indices, which represent markets as a whole, through the investigation of the dynamical behavior of the sub-graph of the LTM members.

Therefore, according to the Reviewer's suggestion, we have changed the presentation of our work in the Introduction in order to make clear from the very beginning what the main message of the paper is.

2. *What I missed is a connection to other topological indicators that might be related to the one presented in the paper (community detection algorithms).*

We thank the Referee for highlighting this point and we agree on her/his observation. Our methodology can be related to other indicators and community detection algorithms. It is undoubtedly that the assessment of the community structure of a network allows us to understand how tight the mutual interconnections among the nodes are, hence providing useful insights on the stability conditions of a system. In this respect, our approach relates to the algorithms employing community detection on empirical correlation matrices (Newman (2010); Fortunato (2010); MacMahon and Garlaschelli (2015)). When applied to financial time series, these approaches have been shown to capture the dynamics of real markets. Related to our work, Kritzman et al. (2011) introduced an indicator, namely the absorption ratio, to estimate the fraction of total market variance explained by a finite number of factors of a principal component analysis applied on several markets and on a rolling basis. They show how a standardized measure of shifts in the absorption ratio can relate to market turbulences. More in general, Bisias et al. (2012) suggest a comprehensive systemic risk assessment in which both micro and macro dimensions of risk should be employed to detect financial stress, emphasizing the importance to evaluate the risk in a time dimension (e.g., procyclicality) and the allocation of risk in a cross-sectional dimension (e.g., common exposures and interlinkages). As suggested by Borio (2010), the most appropriate risk measures differ for these two dimensions, where early-warning or leading indicators are best for addressing the time dimension, while measures of each single entity's contribution to the stability of a financial system are appropriate for the cross-sectional dimension. In our work, the identification of the LTM thus aims at capturing both dimensions and we interpret the LTM time variations as important signs of financial instability which we refer to the presence of sudden critical transitions that have been already widely documented to occur in financial markets during distressed periods (see e.g. De Bandt and Hartmann (2000); Lorenzoni and Werning (2013)).

According to the Reviewer's request, the revised manuscript contains references to the above mentioned approaches that can be related to ours, while a wider discussion is left in the present response due to the space limitation.

3. *In the abstract the sentence: "Our study reveals the emergence of a sub-graph of stocks, among those referring to a benchmark financial index, as the marker of an out-of-equilibrium transition of the underlying system." is hard to understand in the given context and perhaps instead of the term "financial index" the term "market index" is more useful.*

We thank the Reviewer for having raised this point which helps us to better clarify the outcomes of our analysis. We agree with her/his suggestion and, instead of financial index we shall make reference to a market index. Moreover, the above mentioned sentence of the abstract has been rephrased as follows: “Our study reveals how the arising of a sub-graph of stocks from a broader market index represents the signal of an out-of-equilibrium transition of the underlying system”.

4. *What tends to be the industry or market capitalization composition of the LTM stocks? Any pattern there?*

We thank the Reviewer for having raised this point. We have information on market capitalization of the stocks composing the indices, so that we can perform an analysis on such weights to find whether there are patterns in their evolution. We find that apparently no clear evidence emerges from this analysis, meaning that the LTM membership is not heavily correlated with the weights of the stocks composing the corresponding aggregate market index. For instance, for the Asian index that correlation is only 0.038 (p-value 0.047), while for the US market is 0.051 (p-value 0.053). We found that, on average, the weight of the LTM members accounts for the 10% of the market index with peaks of 30%. Below we report the results of the analysis for both Asia and US stock markets.

Figure 1: **Weight analysis for the STOXX Asia/Pacific 600.** The figure reports the weights of the LTM members in the market index composition.

Figure 2: **Weight analysis for the STOXX North America 600.** The figure reports the weights of the LTM members in the market index composition.

5. *It might help for the general reader if you would explain, intuitively following terms that are used in the Introduction section, endogenous instabilities, positive feedbacks, herding behavior.*

We thank the Reviewer for having raised this point related to some terms that may be hard to understand for a general audience. According to her/his suggestion, we have explained the meaning of the above mentioned terms. In particular, in the present response we provide a longer explanation about these concepts, while in the manuscript we shall include a shorter version due to the space limitation.

In general, when talking about endogenous instabilities in financial markets, we refer to the erratic dynamics of markets which is to a large extent of endogenous origin being determined by the trading activity of market agents and not due to the rational processing of exogenous news. Collective effects mediated by imitation penetrate into markets and lead to instabilities. It is well known that being influenced by the behaviour of others seems to be one of the most common human trait, that persists across history, but at the same time this human habit has lead to dramatic effects (see e.g. Brock and Durlauf (2001); Michard and Bouchaud (2005)), including crises and market crashes. Within this context, competition and complexity could be the essential cause of market endogenous instability.

As concerns the notion of positive feedbacks, firstly it is worth to mention that an economic setting is a feedback system, where the dynamics of the aggregate variables depend on individual expectations and vice versa. When talking about a positive feedback system we refer to a

system that reinforces a change in input by responding to a perturbation in the same direction. Putting this definition in the context of financial markets, we may refer to a situation in which if expectations about the future price of a speculative asset are high, the demand for that asset will increase and the realized price will be also high, so that a large cumulative market move may ensue. Finally, broadly speaking, the concept of herding behavior refers to people that are imitating rather than using their own information or making independent decisions. Stated in other words, an investor herds when the knowledge about other investors' allocation decisions influences her/his investment decision, meaning that they use similar investment practices to those previously applied by other investors, even when this is not justified by their information set (see Trueman (1994); Sharma and Bikhchandani (2000)). Accordingly, herding behavior is particularly relevant in the domain of finance, where it has been discussed in relation to the collective irrationality of investors, including stock market bubbles (see Banerjee (1992)).

6. *I found the following sentence unnecessary: Interestingly, if expectations about the future price of a certain asset are high, then the demand for that asset is likely to increase and the realized price will also be high, so that a large cumulative move may ensue, leading to the broken balance between autonomous conducts and peer influence (Preis et al. (2011)).*

We agree with the Reviewer on this point. In the revised version of the manuscript, we have removed the above mentioned sentence. We have only partly utilized the aforementioned sentence raised by the Reviewer in the replay of point 5 to provide an example of the concept of positive feedbacks in financial markets.

7. *In Section 2 you refer to Chen et al., (2012) where you write about “the system approaching a tipping point”, in Chen et al., (2012) their indicator used standard deviation of group components and in your paper absolute auto-covariances are used. Could you elaborate in the paper in what way is your indicator different from that of Chen et al., (2012)? What is the intuitive motive of using these indicators? The technical details and derivations are shown in Chen et al., (2012) and in your SI, but in the text an intuitive explanation would be helpful.*

We thank the Reviewer for this useful comment that allows us to better explain the economic intuition behind the derivation of our indicator. Indeed, the present approach is general and knuckles to a wider category of complex systems (e.g. ecosystems, climate systems, economic and financial systems) characterized by sudden catastrophic shifts during the onset of a crisis phase (see e.g. Scheffer et al. (2001); Dakos et al. (2008); Quax et al. (2013)). A similar technique has been adopted in Chen et al. (2012) who proposed Dynamical Network Biomarkers (DNBs) to detect EWSs for the progression of complex diseases using throughput data. Differently from Chen et al. (2012), who look at the dispersion of the time series near a critical transition, in the detection of EWSs of financial crisis, beside variables co-movement, we are also keen in quantifying the self-similarity of the returns through the consideration of the autocovariance. In doing so, we aim at mapping some behavioral attitudes of market participants such as positive feedbacks and herding behaviors that reverberate in the path of stock prices. Indeed, it has been empirically observed that, on the one hand, in a financial environment where a self-organizing process introduces positive feedbacks (see e.g. Zhou and Sornette (2003)) to the overall system, the future values of the stock price will depend on the

present one, empirically translating into an increased autocovariance of stock returns; on the other hand, herding behavior effects (see Scheffer et al. (2009); Preis et al. (2011); Scheffer et al. (2012); Kefi et al. (2014); Moon and Lu (2015) among others), that lead agents to act collectively without a centralized direction, empirically drive an increase of the correlation (see Dakos et al. (2010)) of such returns. According to this view, financial crisis are specific features which stem from the non linear interactions among investors' decisions (see Sornette (2017)). The revised version of the paper takes into account the aforementioned concepts. In addition, in order to better emphasize the practical usage of our indicator in a simple investment strategy, in the revised manuscript we have also shown the profits and losses obtained by employing the indicator proposed by Chen et al. (2012) and the lower performance with respect to our indicator, which suggests that the behavioral features of market participants captured through the use of the autocovariance are instrumental for anticipating the financial system dynamics.

8. *You use notation N_t^o to denote stocks that are not in N_t^{LTM} but latter you seem to use N_t^o to denote stocks that are not in N_t^H . It does not disturb me, but if I am correct, your notation here is not precise.*

The Reviewer is right in highlighting this point and, following her/his observation, we have modified the notation in the main corpus of the paper and in the Supplementary Information.

9. *The idea that there is a group that becomes more cohesive and distinct from the rest of the network before the system is experiencing a transition is nice. And hopefully useful. I think that you could elaborate on why the approach should work in the first place. Perhaps, when market is driven by changes in one or two sectors, returns from stocks from these sectors get more correlated, more distinct and perhaps persistent. But why only one such group? Have you considered a case with multiple groups?*

We thank the Reviewer for this comment. In the Supplementary Information we have proven that there exists a group of stocks whose dynamics is related to the leading eigenvalue of the Jacobian matrix describing the financial system in a hidden space. For identifying these stocks we have proposed a clustering procedure and an indicator based on returns' correlation and autocovariance. We indeed refer to the LTM as the group with the highest indicator value. This means that such group is the one which drives the whole system to a new equilibrium configuration. For the sake of our analysis we have not discriminate stocks by sector but the LTM could contain stocks derived for different sectors forming a module in the clustering procedure. Here we show that the indicator computed on the LTM provides enough information on the system dynamics and that it is clearly different from the one computed on the module which has the second highest value of such indicator, especially during boom and burst market phases. This results highlights the reason why we employ only one module, that is the one with the highest value of I .

Figure 3: I^{LTM} versus the second highest indicator The figure reports the I^{LTM} dynamics against the pattern of the second highest indicator for the STOXX Asia Pacific 600 case.

10. *The use of DFA pre-filtering for market data seems to be slightly odd, but it might be my background, where I would expect an econometric approach (ARMA-GARCH filtering). This is just an observation of mine.*

We thank the Reviewer for this comment. We decided to rely on a DFA approach due to the presence of an extant literature within complex systems, from which we believe our study may find more interested readers, that relies on DFA-like techniques to detect and study, e.g., volatility clustering, irregular market behaviors and long-range relationships in financial domains (see Muchnik et al. (2009); Lin et al. (2015) among others). Undoubtedly, other econometric approaches can be utilized to extract meaningful signals from the series, as the ones suggested by the Reviewer. For instance, long-memory properties of financial time series have been deeply analyzed by means of ARCH model, its generalization (GARCH), and in many variants which followed later. Specifically, ARCH/GARCH models have shown to exhibit weak persistent behavior especially on long-horizons, where the time correlation disappears and a simple uncorrelated *Itô* process can recover, while stronger long-range correlations can be displayed by the fractionally integrated generalized autoregressive conditional heteroskedasticity (FIGARCH) model (Mantegna and Stanley 1995; Baillie et al. 1996; Engle 2002; Gabaix et al. 2003; Carbone et al. 2004). From a different perspective, the Kitagawa grid and the extended Kalman filtering methods require instead a priori assumptions on the stochastic process and on the probability distribution function of the random variables (Hamilton 1994). In our work we have proposed a basic application of the DFA approach in order to present in a simple way the selection of the series according to the estimated exponents, whose values allow an intuitive relationship with the features of the underlying stochastic process that we also use to propose illustrative variants of the investment strategies (reported in the Appendix).

11. *I recommend to name the axis in Panel A, B of the Figure 1.*

Thanks for the suggestion. We have labelled the axis in Panels A and B of Figure 1.

12. *I would advise to re-phrase the following sentence: “while correlation values of the LTM members indicate the presence of a bunch of stocks that highly co-move among themselves and...”*

According to the Reviewer’s suggestion we have modified the sentence as follows: “while high correlation values of the LTM members indicate the presence of a bunch of stocks having strong synchronized patterns, which deviate from the behavior of the rest of the system”. We thank the Reviewer.

13. *“Changes in the LTM composition is important” - are.*

Thanks for highlighting the typo. We have corrected it.

14. *In Figure 2, LTM and STOXX should have a slightly different color or line style.*

Thanks for the observation, we have changed the lines style of the Figure to improve its readability.

15. *“...the most recent value of the indicator is compared against its empirical distribution computed over the previous working month.” Against previous calendar month or just previous approx. 22 days that correspond to number of trading days in a month. The later seems to be a more usual choice.*

We thank the Reviewer for this observation. He/she is right. Nevertheless we have completely revised the part of the paper devoted to the trading strategy. In particular, in the main text we have inserted a table with the sensitivity analysis of the P&L to changes in the parameters’ values by taking into account different length for the moving windows, which is instrumental for building the indicator distribution. Therefore, in the revised version of the paper we have shown in Figure 3 the P&L obtained by considering a moving window of three trading weeks. The related sentence has been modified as follows: “the most recent value of the indicator is compared against its empirical distribution computed over the previous three trading weeks (15 working days).”

16. *“To detect the direction of the market trend, we exploit information contained in the average return of the LTM members.” Over which period? Also over the past month?*

The Reviewer is right. In the revised version of the manuscript we have clarified that, for detecting the direction of the market trend, we have exploited the last available information about the LTM members returns, i.e. we have used the most recent value of the returns of the stocks composing the LTM, computing the average return among such members. In

particular, in the main text, we have stated that: “To detect the direction of the market trend, we exploit the most recent returns of the LTM members, averaging among such values”.

17. *You report P&L of 60% please annualized this measure.*

Following the Referee’s suggestion we have annualized the P&L performance of our trading strategy. Accordingly, we have modified the corresponding sentence as: “In fact, by assuming transaction costs of 10 basis-points for each portfolio rebalance, we get a PL of about 5.5% per year.”

18. *In the SI you report results from other benchmarks, I think they should be in the main paper if possible. At least, the Buy & Hold. Also you might consider some other model based benchmark strategy that predicts market up and down turns.*

We thank the Reviewer for this useful comment that allows us to better describe our results. According to the raised observation we have exploited the indicator proposed by Chen et al. (2012) to build an alternative investment strategy. Moreover, we have also created a new investment strategy based on simple average correlation of stock returns. Results have been inserted in the revised version of the paper (Figure 3) in which we show that our strategy outperforms, in terms of profits and losses, the alternative strategies. This comparison is useful for justifying the autocovariance as an alternative measure in the composition of the indicator since this measure is related to the behavioral attitudes of market participants and, in particular, it accounts for the presence of positive feedbacks in the financial market. Finally, this exercise is also relevant for justifying the introduction of a clustering approach for discriminating between the patterns of stock sub-groups in order to anticipate the aggregate market dynamic.

References

- Baillie, R. T., T. Bollerslev, and H. O. Mikkelsen, 1996: Fractionally integrated generalized autoregressive conditional heteroskedasticity. *Journal of Econometrics*, **74** (1), 3–30.
- Banerjee, A. V., 1992: A simple model of herd behavior. *The Quarterly Journal of Economics*, **107** (3), 797–817.
- Bisias, D., M. Flood, A. W. Lo, and S. Valavanis, 2012: A survey of systemic risk analytics. *Annu. Rev. Financ. Econ.*, **4** (1), 255–296.
- Borio, C., 2010: Implementing a macroprudential framework: Blending boldness and realism. *Bank for International Settlements*.
- Brock, W. A., and S. N. Durlauf, 2001: Discrete choice with social interactions. *The Review of Economic Studies*, **68** (2), 235–260.
- Carbone, A., G. Castelli, and H. E. Stanley, 2004: Time-dependent hurst exponent in financial time series. *Physica A: Statistical Mechanics and its Applications*, **344** (1-2), 267–271.

- Chen, L., R. Liu, Z.-P. Liu, M. Li, and K. Aihara, 2012: Detecting early-warning signals for sudden deterioration of complex diseases by dynamical network biomarkers. *Scientific Reports*, **2**, 342.
- Dakos, V., M. Scheffer, E. H. van Nes, V. Brovkin, V. Petoukhov, and H. Held, 2008: Slowing down as an early warning signal for abrupt climate change. *Proceedings of the National Academy of Sciences*, **105** (38), 14 308–14 312.
- Dakos, V., E. H. van Nes, R. Donangelo, H. Fort, and M. Scheffer, 2010: Spatial correlation as leading indicator of catastrophic shifts. *Theoretical Ecology*, **3** (3), 163–174.
- De Bandt, O., and P. Hartmann, 2000: Systemic risk: a survey.
- Engle, R., 2002: New frontiers for arch models. *Journal of Applied Econometrics*, **17** (5), 425–446.
- Fortunato, S., 2010: Community detection in graphs. *Physics Reports*, **486** (3-5), 75–174.
- Gabaix, X., P. Gopikrishnan, V. Plerou, and H. E. Stanley, 2003: A theory of power-law distributions in financial market fluctuations. *Nature*, **423** (6937), 267.
- Hamilton, J. D., 1994: State-space models. *Handbook of econometrics*, **4**, 3039–3080.
- Kefi, S., and Coauthors, 2014: Early warning signals of ecological transitions: methods for spatial patterns. *PloS one*, **9** (3), e92097.
- Kritzman, M., Y. Li, S. Page, and R. Rigobon, 2011: Principal components as a measure of systemic risk. *The Journal of Portfolio Management*, **37** (4), 112–126.
- Lin, A., H. Ma, and P. Shang, 2015: The scaling properties of stock markets based on modified multiscale multifractal detrended fluctuation analysis. *Physica A: Statistical Mechanics and its Applications*, **436**, 525–537.
- Lorenzoni, G., and I. Werning, 2013: Slow moving debt crises. Tech. rep., National Bureau of Economic Research.
- MacMahon, M., and D. Garlaschelli, 2015: Community detection for correlation matrices. *Physical Review X*, **5** (2), 021006.
- Mantegna, R. N., and H. E. Stanley, 1995: Scaling behaviour in the dynamics of an economic index. *Nature*, **376** (6535), 46.
- Michard, Q., and J.-P. Bouchaud, 2005: Theory of collective opinion shifts: from smooth trends to abrupt swings. *The European Physical Journal B-Condensed Matter and Complex Systems*, **47** (1), 151–159.
- Moon, H., and T.-C. Lu, 2015: Network catastrophe: self-organized patterns reveal both the instability and the structure of complex networks. *Scientific Reports*, **5**, 9450.
- Muchnik, L., A. Bunde, and S. Havlin, 2009: Long term memory in extreme returns of financial time series. *Physica A: Statistical Mechanics and its Applications*, **388** (19), 4145–4150.

- Newman, M., 2010: *Networks: An introduction*. Oxford University Press.
- Preis, T., J. J. Schneider, and H. E. Stanley, 2011: Switching processes in financial markets. *Proceedings of the National Academy of Sciences*.
- Quax, R., D. Kandhai, and P. M. Sloom, 2013: Information dissipation as an early-warning signal for the lehman brothers collapse in financial time series. *Scientific reports*, **3**, 1898.
- Scheffer, M., S. Carpenter, J. A. Foley, C. Folke, and B. Walker, 2001: Catastrophic shifts in ecosystems. *Nature*, **413 (6856)**, 591.
- Scheffer, M., and Coauthors, 2009: Early-warning signals for critical transitions. *Nature*, **461 (7260)**, 53.
- Scheffer, M., and Coauthors, 2012: Anticipating critical transitions. *Science*, **338 (6105)**, 344–348.
- Sharma, M. S., and S. Bikhchandani, 2000: *Herd behavior in financial markets: A review*. 0-48, International Monetary Fund.
- Sornette, D., 2017: *Why stock markets crash: critical events in complex financial systems*. Princeton University Press.
- Trueman, B., 1994: Analyst forecasts and herding behavior. *The Review of Financial Studies*, **7 (1)**, 97–124.
- Zhou, W.-X., and D. Sornette, 2003: 2000–2003 real estate bubble in the uk but not in the usa. *Physica A: Statistical Mechanics and its Applications*, **329 (1-2)**, 249–263.

REVIEWERS' COMMENTS:

Reviewer #3 (Remarks to the Author):

This is an improved version of the manuscript. Several of the issues I have raised are now clarified. Authors also provide lengthy discussion to some of the issues raised by both reviewers. Overall I do acknowledge that some of the points are simply matter of personal preference, therefore if I do not see anything particularly wrong with their approach I accept their response. For example, author's test several trading strategies and usually data-snooping robust tests could be employed (e.g. Hansen's et al., 2010, The Model Confidence Set, Econometrica), but on the other hand not that many trading strategies are employed so that does not disturb me that much. Also, authors could put a few more words on how they perform the bootstrap. They are silent about it, while it is unclear whether the bootstrap samples resemble the underlying sample (e.g. volatility clustering of returns?). I do not think that it is going to do much, but I missed that information. To conclude, I therefore lean towards acceptance.

This is an improved version of the manuscript. Several of issues I have raised are now clarified. Authors also provide lengthy Perhaps, authors could put a few more words on how they perform the bootstrap. They are silent about it, while it is unclear whether the bootstrap samples resemble the underlying sample (e.g. volatility clustering of returns?). I do not think that it is going to do much, but I missed that information. To conclude, I therefore lean towards acceptance.

Reviewer 3

We are grateful for the Reviewer's comments and suggestions, which we used to improve the paper. We reproduce the comments below and reply to the points raised by the Reviewer:

1. *This is an improved version of the manuscript. Several of the issues I have raised are now clarified. Authors also provide lengthy discussion to some of the issues raised by both reviewers. Overall I do acknowledge that some of the points are simply matter of personal preference, therefore if I do not see anything particularly wrong with their approach I accept their response. For example, authors test several trading strategies and usually data-snooping robust tests could be employed (e.g. Hansen et al., 2010, The Model Confidence Set, Econometrica), but on the other hand not that many trading strategies are employed so that does not disturb me that much. Also, authors could put a few more words on how they perform the bootstrap. They are silent about it, while it is unclear whether the bootstrap samples resemble the underlying sample (e.g. volatility clustering of returns?). I do not think that it is going to do much, but I missed that information.*

We thank the Reviewer for this useful comment that allows us to better describe our bootstrapping strategy.

We use bootstrapping to estimate the confidence intervals of the indicator from sample data. Our bootstrapping procedure involves random permutation of data points for each time series at each step of the moving window (see also (1)). The number of elements in each bootstrap sample is then equal to the number of elements in the original data set. This allows us to look at the distributions of outcome and to compute confidence limits. In other words, we obtain bootstrap samples by randomly shuffling the stock returns; in this way we test whether the temporal ordering of the market returns matter in the computation of the I^{LTM} . Error bounds are computed by performing 500 bootstraps re-sampling by randomly permuting stocks' returns. Every bootstrap sample allows acquiring an estimate of I_t^{LTM} which is used to compute the distribution of the reshuffled indicator and to estimate the error bound as the 5-95 percentiles of such distribution.

References

- [1] Stengos, T. *Nonparametric Econometric Methods and Applications* (Multidisciplinary Digital Publishing Institute, 2019).